# Confidence control for efficient behaviour in dynamic environments

**Tarryn Balsdon** [1,2] ✉ **& Marios G. Philiastides** [1]

Signatures of confidence emerge during decision-making, implying confidence may be of functional importance to decision processes themselves. We formulate an extension of sequential sampling models of decision-making in which confidence is used online to actively moderate the quality and quantity of evidence accumulated for decisions. The benefit of this model is that it can respond to dynamic changes in sensory evidence quality. We highlight this feature by designing a dynamic sensory environment where evidence quality can be smoothly adapted within the timeframe of a single decision. Our model with confidence control offers a superior description of human behaviour in this environment, compared to sequential sampling models without confidence control. Using multivariate decoding of electroencephalography (EEG), we uncover EEG correlates of the model's latent processes, and show stronger EEG-derived confidence control is associated with faster, more accurate decisions. These results support a neurobiologically plausible framework featuring confidence as an active control mechanism for improving behavioural efficiency.

Feelings of confidence in decision accuracy are thought to reflect a metacognitive evaluation of the quality and quantity of evidence underlying decisions[1,2]. Confidence has been extensively studied as a post-decision reflection[3] which is important for communicating about past decisions[4] and improving future decisions[5–7]. However, there is increasing experimental support for signatures of confidence emerging online, during decision-making[8–11], suggesting confidence could serve an active role prior to decision commitment. Indeed, there is a growing endorsement for a role of confidence in adjudicating how much evidence is required to commit to decisions[12–15], consistent with earlier proposals implicating a belief about obtaining a reward (cost-benefit trade-off[16,17]). Here, we test a framework in which confidence does not only control the quantity of decision evidence, but is used to actively improve evidence quality. Confidence thus manifests as a central control mechanism for moderating behavioural efficiency, that is, maximising precision given the constraints of time and effort.

We implement this role of online confidence in controlling the quality and quantity of decision evidence using a double-integration framework[18,19]. Dominant theories of decision formation rely on single-integration frameworks, where decision evidence is accumulated over time until a bound is reached[20–22]. With only a single stage of evidence accumulation (single-integration framework), the relative (to the starting point) placement of the bound is the only mechanism for controlling behavioural efficiency. These bound adjustments, including confidence-controlled variants[12,13], have been previously used to describe the speed-accuracy trade-off (SAT[23]) and decision urgency[24] (in addition to evidence-boosting urgency signals[25–27]). Instead, in our double-integration framework, the primary source of evidence accumulation remains unbounded, with the bound placed on the evidence re-integrated by a secondary accumulator. This secondary accumulator has a "leaky memory" and uses this property to consider past evidence in the accumulated signal, rather than just its current state. Importantly, this property serves to modulate both the quality and quantity of decision evidence, which here we implement as being controlled by online confidence computed from the primary accumulator.

The inclusion of a secondary accumulator in the double-integration framework was motivated by its neurobiological

[1]School of Psychology and Neuroscience, University of Glasgow, Glasgow, United Kingdom. [2]Laboratory of Perceptual Systems, DEC, ENS, PSL University, CNRS (UMR 8248), Paris, France. ✉e-mail: tarryn.balsdon@ens.fr

plausibility: this secondary accumulation is proposed to take place at the level of motor processes, to explain the build-up of motor activity prior to decision commitment[27–29], and evidence for a more active role of these motor processes in decision-making[30–33]. In addition, the primary accumulator is left unbounded, to account for how neural signatures of (primary) evidence accumulation can appear to terminate at different bounds, depending on the quality of evidence[34–36] and also persist beyond decision commitment[8,37,38]. We have already shown how controlling the "leaky memory" of the secondary accumulation offers a better characterisation of SAT compared to bound adjustments[19], and how this provides a more holistic explanation of observed electroencephalography[18] (EEG). This framework has also been used to explain partial electromyography activity profiles[39]. Here, we seek to investigate the role of online confidence in controlling the double-integration of evidence and demonstrate how perceptual, inference, metacognitive, and motor functions – classically studied as distinct processes – are effectively working in concert to dynamically orchestrate efficient behaviour.

While the role of confidence in controlling decision processes makes theoretical sense, most current computational models of decision formation sufficiently explain behaviour without a prominent need for confidence control. The role of online confidence control, however, would be most pronounced in situations where high-quality evidence may not last, and low-quality evidence may improve (situations more like natural environments). In these scenarios, the observer would benefit from adapting decision processes online[17] to capitalise on early high-quality evidence, or down-weight early low-quality evidence in case better evidence follows. While previous experiments have manipulated the volatility of sensory evidence across trials (via random variations of evidence quality[40]), or created systematic but predictable changes in sensory evidence quality within blocks of trials[41], behaviour in these environments can

be explained by offline adaptations to sensory statistics as opposed to online control.

Here, we designed a dynamic sensory environment in which evidence changes in a systematic but unpredictable manner during the course of individual decisions. In this context, efficient behaviour requires online control of decision processes in response to temporally evolving stimulus evidence. We show that a classic single-integration model fails to capture human behaviour in this environment. In contrast, we show that our double-integration model with confidence control can reliably capture behaviour and that the dynamics of these two integration processes can be mapped onto distinct spatiotemporal EEG signatures. Moreover, we show that endogenous variability in a separate EEG-derived signature of confidence control could be used to predict intraindividual behavioural efficiency. Together, our findings offer evidence in support of our double-integration framework and the role of online confidence control in arbitrating behavioural efficiency.

## Results

Participants were asked to perform a fine-grained left/right (from vertical) direction discrimination of moving dot stimuli, and rate their confidence that they were correct (Fig. 1a). The stimuli provided dynamic evidence for the decision, where the quality of the sensory evidence could increase or decrease within the first second of stimulus presentation (within the timeframe of individual decisions). The quality of the sensory evidence was controlled by manipulating the mean and variance of the circular Gaussian (von Mises) distributions from which the dot directions were sampled. There were nine conditions (Fig. 1b): two with increasing evidence quality (increasing the mean or decreasing the variance); two with decreasing evidence quality (decreasing mean or increasing variance); three with stable evidence quality (low/moderate/high); and two with moderate

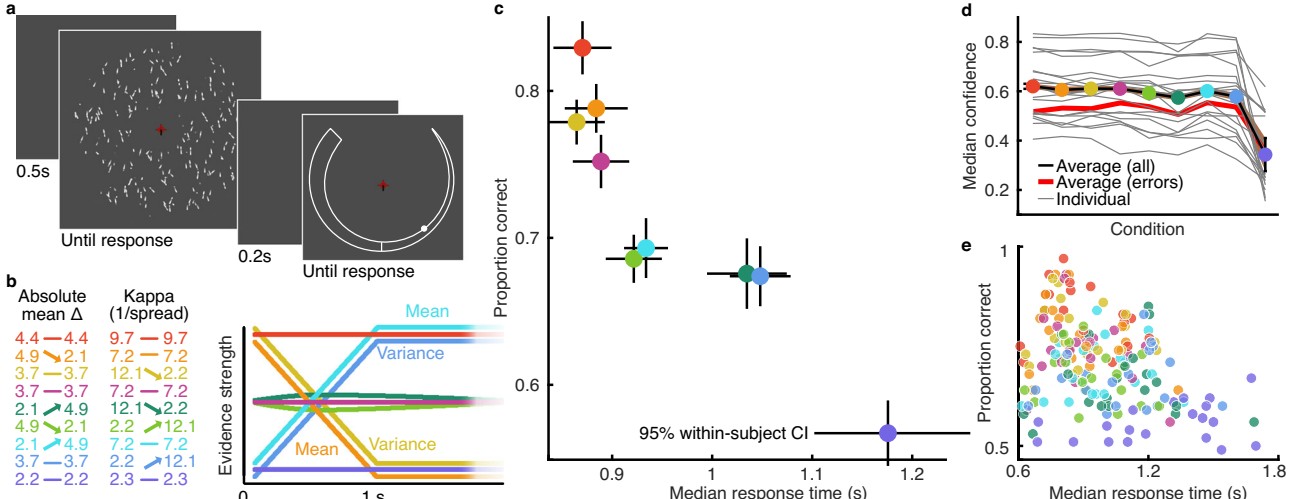

**Fig. 1 | Behaviour in a dynamic sensory environment. a** On each trial, the observer was presented with a display of moving dots, presented until they entered their response (up to 2 s). After 200 ms, observers were cued to make a confidence rating, entered after unlimited time by holding a response key to move the circular marker around the annulus until it reached their desired confidence (thicker annulus means higher confidence; it was possible to make more than one revolution to return to lower confidence). **b** The quality of the sensory evidence was manipulated according to nine conditions created by adjusting the parameters of the circular Gaussian (von Mises) distribution from which the dot directions were sampled. The table to the left shows the parameter values in each condition: the mean difference (Δ) from vertical (degrees) and the Kappa parameter, which controls the inverse spread of the distribution (arrows highlight the change in terms of evidence strength, which is also depicted to the right). In four conditions, the

evidence was systematically increased or decreased over the first second of stimulus presentation (by increasing/decreasing the mean or the variance). In three conditions, the evidence was completely stable. Two additional conditions of moderate evidence were created by changing the mean and variance in opposite directions. **c** Proportion correct by median response time in each condition (colours correspond to conditions in b) averaged across 20 participants (100 trials per participant), lines show 95% within-subjects confidence intervals. **d** Median confidence in each condition, thin grey lines show individual participants, the thick red line shows the average median confidence of incorrect trials, the black line shows the average of all trials, and the shaded region shows 95% within-subjects confidence (barely thicker than a black line). **e** Proportion correct by median response time in each condition (colours correspond to conditions in (**b**) for individual participants (100 trials per condition).

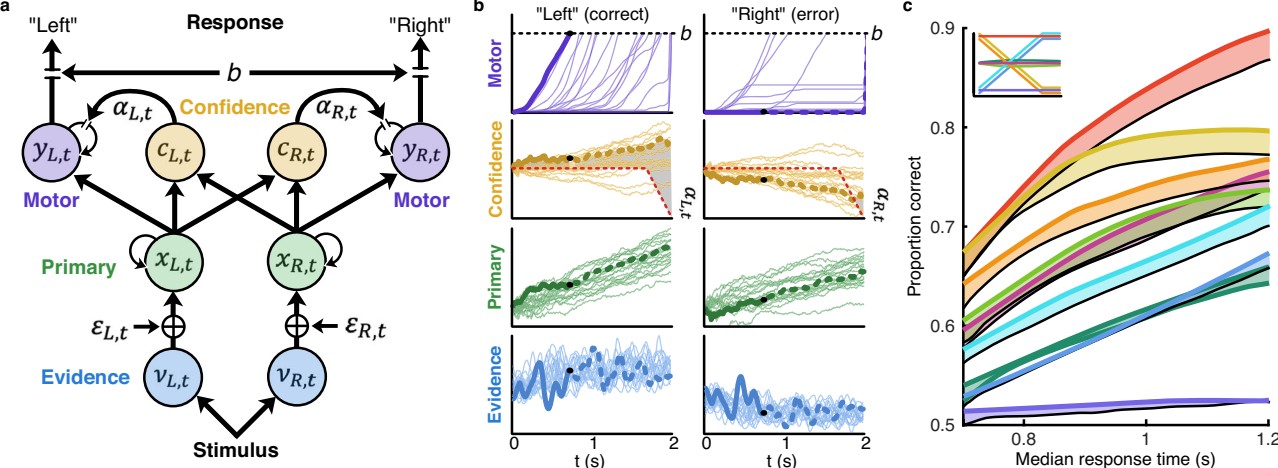

**Fig. 2 | Double integration framework. a** Visualisation of the double-integration model with confidence control. Noisy estimates of the stimulus evidence ($v$) are integrated into the primary accumulator ($x$). Classic single-integration models propose a bound set on the primary accumulator to determine the response. Here the primary accumulator remains unbounded and informs both confidence ($c$) and the secondary motor accumulator ($y$), which re-integrates the already integrated evidence ($x$) with some leakage ($\alpha$) controlled by confidence. The bound ($b$) on the motor accumulator determines the response, while the primary accumulator could continue to accumulate additional evidence, for example, to inform post-decision confidence. **b** Example traces simulated from 20 example stimuli from the decreasing variance condition. One stimulus is highlighted in bold, decision commitment would have occurred at the black dot (continued in dashed line for demonstration). The grey-shaded region on the confidence plot shows the leakage, which is proportional to confidence greater than 0.5. The best-fitting model incorporated a temporal decay in this lower confidence bound (red dashed line) such that over time, leakage can increase based on lower confidence, simulating an urgency effect. **c** Proportion correct by median response time simulated using the double-integration framework (coloured lines corresponding to conditions in Fig. 1b, inset, average of 1000 simulations), or terminating the decision with a bound on the primary accumulator (black lines). Shaded regions illustrate the additional benefit of double-integration with confidence control, which improves both the speed and accuracy of decisions.

evidence quality created by manipulating the mean and variance in opposite directions (see "Methods" for details). Stimuli from these conditions were presented in intermixed order, thereby limiting the participants' ability to predict how evidence quality might change based on the initial evidence. Participants were instructed to respond as quickly and accurately as possible while the stimulus presentation continued until their response (or up to 2 s).

## Decision processes are modulated by sensory evidence dynamics

The average proportion correct by median response time (RT) is plotted in Fig. 1c (individuals in Fig. 1e), showing participants make fast, accurate responses to early high-quality evidence whilst also slowing down when faced with early low-quality evidence. There was a significant effect of stimulus condition on discrimination performance (d', repeated measures ANOVA, $F_{(8152)} = 47.83$; $p < 0.001$), response times ($F_{(8152)} = 40.14$; $p < 0.001$), and confidence (Fig. 1d; $F_{(8152)} = 70.91$; $p < 0.001$). These effects are driven by differences between most pairs of conditions (paired $t$ tests in Supplementary Tables S1–S3).

## Double integration with online confidence control

We formulated an extension of the Leaky Integrating Threshold model (LIT[19]) to account for online modulation of decision processes via metacognitive confidence control. The LIT model employs a double-integration framework, which was initially motivated by neurobiological plausibility: the primary accumulated evidence is re-integrated by a secondary, motor accumulator, which triggers the behavioural response at its own bound. This accounts for the build-up of motor activity prior to decision commitment and formalises the more active role of motor processes in decision-making[30,31]. This framework also allows for flexible online modulation of the evidence accumulation rate, via leakage in the motor accumulator. In previous work, we showed how the model can capture changes in the speed-accuracy trade-off via changes in a static leakage parameter[18,19] (leakage was constant within decisions; henceforth referred to as the static LIT

model). Here we formulate an extension by proposing that motor leakage is controlled by confidence, which is computed online from the (unbounded) primary accumulator. Critically, this means online confidence control is implemented without circularity since the evidence for confidence is decoupled from the final evidence for the decision.

The double-integration framework is visualised in Fig. 2a, with equations provided in the Methods. In the context of the current task, the observer takes the perceptual evidence ($v$) as the log probability of leftward vs rightward dot directions based on the presented stimulus. This perceptual evidence is disrupted by additive Gaussian noise ($\varepsilon$), with zero mean, and standard deviation modulated by a free parameter. The momentary perceptual evidence is integrated over time at the primary accumulator ($x$). Accumulating the log probabilities amounts to a sequential probability ratio test, the Bayes optimal task solution. Classic single-integration models propose that a bound is placed on this primary accumulated evidence to dictate the response. This provides the optimal strategy in the case that the observer is unaware of the noise ($\varepsilon$) affecting their representation of the decision evidence. However, with access to some representation of uncertainty, the observer could improve decision efficiency via a secondary evidence integration process (Fig. 2c).

Online metacognitive confidence signals could provide this representation of uncertainty. In this framework, confidence ($c$) is computed based on the primary accumulated evidence and used to moderate the leakage ($\alpha$) in the secondary, motor accumulator ($y$). The motor accumulator re-integrates the already integrated primary evidence. This is an essential difference from previous models with multiple stages, which suggest evidence is merely passed along without re-integration. The leakage in the motor accumulator controls the memory of previous integrated states and thus acts as a smoothing filter (exponential moving average) to actively improve evidence quality (smooth out temporally independent noise) at the same time as slowing down the rate at which the motor accumulated evidence reaches the decision bound ($b$). In previous formulations (the static

LIT), this smoothing parameter was static over time, and we have demonstrated how modulating the smoothing parameter across speed-accuracy trade-off conditions can better account for behaviour and neural signals (EEG) compared to modulating the bound[18,19]. With confidence modulating the leakage in the motor accumulator, the amount of smoothing now becomes a dynamic variable and corresponds to current certainty conditioned on the provisional choices. With low confidence, there is more smoothing, which slows down the rate at which the motor evidence reaches the bound. As confidence increases, smoothing decreases, and the motor evidence speeds toward the bound. Figure 2b shows how these signals evolve during decision-making for 20 example stimuli with increasing evidence strength (decreasing variance condition). In this way, the double-integration framework describes the flow of information over four stages of processing (perceptual, inference, metacognitive, and motor). However, it is elegant in requiring only one additional parameter compared to the traditional single-integration framework (see "Methods" for details).

Importantly, this framework also has the potential to account for post-decision confidence. Confidence is computed from the primary accumulator, which remains unbounded and is allowed to persist past the response. This captures how confidence can incorporate additional evidence accumulated after decision commitment (without having to propose a double-bound, as in single-integration frameworks[37,38]), although without specifying a read-out mechanism. Moreover, the smoothing in the motor accumulator suggests the decision is based on evidence affected by less noise than was present in the primary accumulator, explaining how confidence judgements reflect more intrinsic noise than decisions[42] and accounting for the dissociation between confidence and decision sensitivity[8].

## Computational evidence for online confidence control

Our main hypothesis was that incorporating confidence control in the double-integration framework would provide a better description of participant behaviour than the classic single-integration model. The classic single-integration model dramatically failed to capture the pattern of behaviour across intermixed stimulus conditions (Fig. 3a). The model was unable to describe both how participants could be faster and more accurate with early high-quality evidence while also slowing down with early low-quality evidence. This failure is in part due to the fact that the model has to implement a single decision bound across the intermixed stimulus conditions, and in part due to constraining all models to use the evidence from the presented stimuli (a constraint validated by a separate GLM analysis, Supplementary Fig. S1). This second constraint can be somewhat ameliorated by adding a bottom-up weighting of the evidence as it is accumulated (Fig. 3b), to weaken the influence of more variable stimulus evidence (an overcompensation in the computation of decision evidence; see Eqs. 1–3, 11 in "Methods"). While this bottom-up weighting improves the spread of response times across conditions, the model overestimates performance, and the additional parameter used to implement the weighting is not parsimonious (bottom-up weighting – classic $\sum \Delta BIC = 38.14$; the model also suffers a lack of neurobiological plausibility; see Supplementary Fig. S6).

Similar model failures were also apparent for variants of the single-integration models with time-varying bounds, and for a static LIT model (as implemented in previous work[18,19]) where double-integration occurs with a smoothing parameter that is stable over time (see Supplementary Figs. S2, S6). Describing behaviour in this dynamic sensory environment requires a model where decision processes can react to the unpredictable transitions in the evidence quality online (Supplementary Fig. S6), which is provided by the double-integration model with online confidence control (Fig. 3c; and would otherwise require effective stimulus-time-dependent bounds on the primary evidence[13,14], Supplementary Fig. S2).

Formal model comparison suggested the double-integration model with online confidence control proved superior to the classic model (Fig. 3c; $\sum BIC_{classic} = 8.33 \times 10^4$, $\sum BIC_{confcontrol} = 8.25 \times 10^4$, $\sum \Delta BIC = 842.61$, protected exceedance probability = 0.981; see Supplementary Table S4 for a summary of parameters and fit statistics) and the classic model with bottom-up weighting ($\sum BIC_{bottom-up} = 8.34 \times 10^4$, $\sum \Delta BIC = 880.47$, protected exceedance probability = 0.989). The quality of the fit of the confidence control model can be appreciated from the full accuracy-RT distributions shown in Fig. 3d, as well as the comparison of individual simulated vs participant proportion correct and response times in Fig. 3e.

As additional evidence of the superiority of the double-integration framework, we were also able to predict post-decision confidence ratings from the model fit only to choices and response times, without additional parameters. For each trial, we simulated 1000 noisy instances of the double-integration process, using the parameters fit to each participant, and took the median of the instances consistent with the participant's choice and response time on that trial. This provided a trial-wise estimate of the dynamic primary and motor accumulated evidence, as well as the online confidence computed from the primary accumulator. The final state of the model-simulated online confidence for the chosen response predicted observers' post-decision confidence ratings with $z = 0.41$ (average Fisher transformed correlation; range, [0.0604, 0.5634]; $t$ test against 0, t(18) = 12.425, $p = 1.45 \times 10^{-10}$) and captured the pattern of confidence across decision accuracy and response time (Fig. 3f). This validates the claim that post-decision confidence can be accounted for within the double-integration framework, and shows the latent model variable is related to participants' explicitly reported confidence following their decisions.

## Neural mechanisms of confidence control

The double-integration model was motivated by neurobiological plausibility but provides a numerical description of the processes underlying behaviour that is essentially agnostic to the neural mechanisms implementing these processes. We hypothesised that the primary accumulator corresponds to the accumulation of evidence in the associative (parietal) cortex (as with single-integration models[43,44]). The motor accumulator we hypothesised would be realised by the motor processes corresponding to the effector (in this case, the right hand). We have already demonstrated that the relationship between associative and motor electroencephalography (EEG) components in speed-accuracy trade-off contexts maps well to the predictions of the double-integration framework[18]. Here, we use a different analysis to obtain appreciably similar results (Supplementary Fig. S4). As described above, for each trial, we took traces of primary and motor evidence accumulation based on the median of simulated traces that were consistent with the choice and response time (out of 1000 simulated instances of double-integration using the participants' fitted parameters). Due to the overall ramp profile in both accumulators, the raw primary and motor accumulators were strongly correlated (average Fisher transformed correlation $z = 1.02$) which would disrupt our ability to isolate separable EEG components simultaneously associated with these two accumulators. However, the derivatives (slopes, Fig. 4a) show a weaker correlation ($z = 0.23$), as they distinguish the linear vs exponential profiles of the primary vs motor accumulators (respectively).

We used a decoding analysis to trace the neural representation of simulated primary and motor accumulated evidence traces: An inverted multiple linear encoding model (based on multiple linear regression, see "Methods"), was used to maximally disambiguate the EEG sources contributing to primary and motor accumulation. The inverted encoding model identifies weights on EEG sensors that best predict the simulated evidence derivatives (both primary and motor) from the derivative of the EEG signal amplitude in the 400 ms leading to the

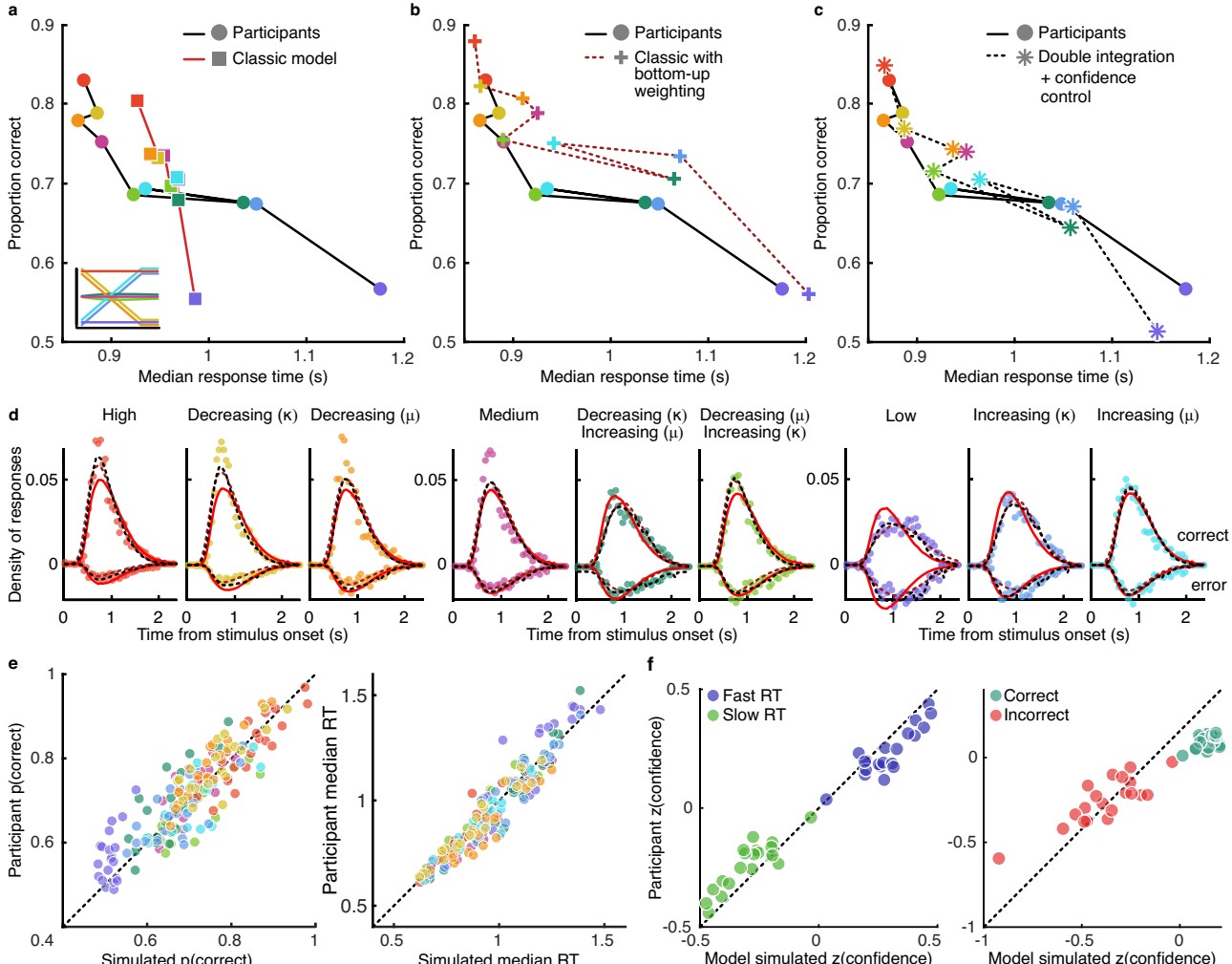

**Fig. 3 | Computational modelling. a** Participant behaviour (circular markers, black line) compared to that predicted by the classic single-integration model (square, red line). Colours correspond to Fig. 1b, inset. **b** Participant behaviour (circular markers, solid line) compared to the classic model with a bottom-up weighting of evidence accumulated (cross markers, red dashed line). **c** Participant behaviour (circular markers, solid line) compared to the double-integration model with online confidence control (asterisk markers, dashed line). **d** Probability density of responses by response time. Correct responses are shown upward of the 0-point, and incorrect, downward. Markers show participants, the red solid line shows the

prediction of the classic model, the red dashed line shows the prediction of the classic model with bottom-up weighting, and the black dashed line shows the prediction of the confidence control model. **e** Participant proportion correct (left) and median response time (right) across stimulus conditions, compared to that predicted by simulating the confidence control model using each participant's fitted parameters. **f** Participants' confidence (z-scored) by confidence predicted from the model simulated confidence (using parameters fit to choices and response times but not confidence), across trials split by median response time (fast RT vs slow RT, left) and correct/incorrect discrimination decisions (right).

response (see "Methods"). The precision, assessed by taking the correlation between the EEG predicted evidence and the model simulated evidence, was not high, but significantly above chance across participants (for the primary accumulator, mean $z = 0.024$, t(18) = 5.90, $p = 1.10 \times 10^{-5}$; for the motor accumulator, mean $z = 0.086$, t(18) = 9.07, $p = 2.47 \times 10^{-8}$). The overall shape of the EEG-predicted evidence accumulation profiles and those described by the model are well matched (Fig. 4a), supporting the existence of these two distinct (linear vs quadratic) accumulation profiles in terms of neural correlates.

The central purpose of this analysis was to assess the neurobiological validity of the proposed computational mechanisms, which can be appreciated from the topography of the sensor weights shown in Fig. 4b: the motor accumulator most strongly relies on EEG signals over contralateral motor cortex; the primary accumulator is associated with topography comparable to a central parietal positivity signal, previously associated with evidence accumulation[44–46] (note that a more symmetric topography emerges when performing the analysis

separately for each accumulator, Supplementary Fig. S4). We consider this strong evidence for the neurobiological validity of the double-integration framework, where the appropriate topography emerges based on single-trial estimations of the two accumulation signals from the computational model that is agnostic to the relevant neural signatures.

We performed a separate analysis to decode post-decision confidence ratings from the EEG signals following the perceptual decision. The resulting topography is shown in Fig. 4c. We then applied these sensor weights from decoding post-decision confidence to the signals just prior to the response to generate an EEG-predicted confidence signal during decision-making. The prediction of the computational model is that these online confidence signals are used to control the smoothing of the motor accumulator, such that increased confidence decreases smoothing and drives faster accumulation to the motor bound. Greater online confidence should, therefore, cause a faster ramp-up in the motor accumulator signal. Indeed, we found significant evidence that the EEG online confidence signals (based on decoding

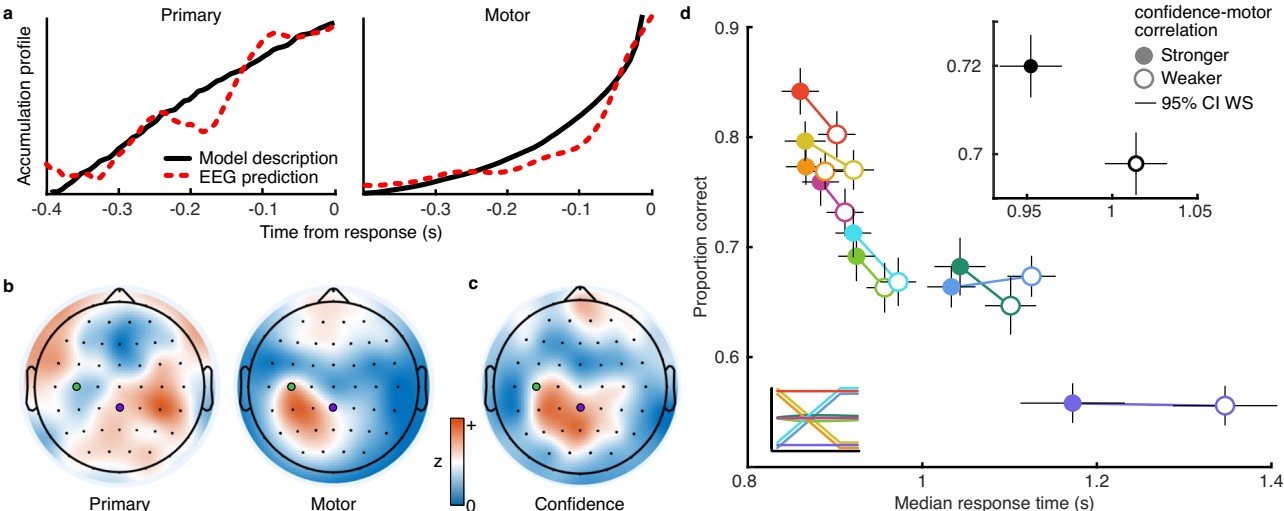

**Fig. 4 | Neural signatures of confidence control. a** Accumulation profile (a derivative of accumulated evidence) of the primary (left) and motor (right) accumulators from simulating the model fit to participants (black) and those predicted from the EEG inverted encoding model (red dashed line). **b** Topography of the EEG encoding of primary (left) and motor (right) evidence accumulation (Fisher transformed precision increases in orange, white marks the half-maximum). The green marker shows sensor C3 (commonly associated with right-hand motor responses), and the purple marker shows sensor CPz (over which the central parietal positivity signal is centred). **c** Topography of the EEG representation of post-decision confidence (same format as **b**). **d** Proportion correct by median response time for trials split by the correlation between EEG-predicted online confidence and EEG-predicted motor accumulation ramp (median split; 50 trials for each of 20 participants per marker). Trials with a stronger correlation (filled) corresponded to faster, more accurate responses in each stimulus condition (coloured as in Fig. 1b, inset), and averaged across conditions (black inset). Error bars show 95% within-subject confidence intervals on the difference between stronger and weaker correlated trials.

post-decision confidence judgements) granger-cause ramp-up in the EEG motor accumulator signals (based on decoding the model predicted accumulation derivative) within each participant (leave-one-out Granger Causality test; minimum $\chi^2 = 477.46$, all $p < 0.001$, with best fitting vector autoregressive models including lags between 272–784 ms). These neural signals, therefore, offer additional support for the hypothesis that confidence is used online to control decision processes.

As an exploratory analysis, we examined how these neural signatures of online confidence control relate to behavioural efficiency. We measured the endogenous variability of the strength of confidence control using the within-trial correlation between the EEG-predicted online confidence and the ramp-up of the EEG-predicted motor accumulator, in the 200 ms prior to the response (where the ramp is most pronounced, prior to which, the smaller ramp makes the analysis more vulnerable to noise). Trials with stronger correlation predict tighter confidence control, which is integral for behavioural efficiency under the double-integration framework (Fig. 2c). We took a median split based on the EEG-predicted confidence-motor correlation, within subjects and within conditions. Trials with stronger EEG-predicted confidence control showed faster, more accurate responses (Fig. 4d; effect on discrimination performance: F(1,19) = 11.19, $p = 0.0034$; effect on response time: F(1,19) = 12.15, $p = 0.0025$). EEG-predicted confidence control is associated with increased behavioural efficiency.

## Discussion

These results provide behavioural and neural evidence for the important role of online confidence control implemented in our proposed double-integration framework. This role of online confidence control is amplified in dynamic stimulus environments, where it is used to modulate decision processes according to unpredictable transitions in stimulus evidence quality. This serves not only to increase behavioural efficiency in dynamic environments, as in our task, but also to improve the speed and accuracy of individual decisions (Fig. 2c and Fig. 4d). The neurobiological validity of the double-integration framework is supported by the spatiotemporal properties of distinct EEG

signals (Fig. 4a, b) that align with the proposed role of associative and motor cortex in primary and motor evidence accumulation. The coordination of these processes functionally contributes to behavioural efficiency, where weaker confidence control of EEG motor accumulation leads to slower, less accurate responses (Fig. 4d). The neurobiological plausibility of this framework, specifically in describing the interaction of several subsystems, will provide an advantage in generating testable predictions across multiple neuroimaging modalities, and across different species of decision-makers.

The proposed framework sets out several shifts in current paradigms of decision-making. First, and most prominently, online confidence actively modulates decision processes to improve current behaviour, as opposed to being a passive post-decision evaluation only useful for improving future behaviour. Second, confidence control moderates not only the quantity of evidence accumulated (as provided by decision boundary shifts) but also the evidence quality (via the smoothing induced by motor leakage). Third, as suggested in previous work and formalised here, motor processes play an active role in decision formation[30,31], ultimately determining the choice at the motor bound, which could be conceptualised as the level of cortical activity sufficient to propel excitation down the spinal cord to the behavioural effector[39]. Fourth, motor leakage is a critical component in modulating the implicit speed and accuracy of decisions in dynamic sensory environments, operating to smooth re-integrated evidence and slow down the rate of evidence build-up. We propose that this motor leakage might be moderated by the arbitration of two endogenous signals, confidence (primarily pushing for accuracy and value) and urgency (primarily pushing for speed and reduced effort), which could be integrated into subcortical structures implementing motor (dis) inhibition[47,48]. In our view, the nature of this arbitration is a prominent component to be examined in future work.

Another contribution of this work is in describing the formation of confidence. We show that the EEG signatures of post-decision confidence can be used to estimate online confidence in a way that is meaningful for behaviour. Moreover, the final estimate of online confidence from the model fit to choices and response times provided

a good prediction of post-decision confidence responses. The model formalises confidence as relying on primary accumulated evidence, prior to the re-integration at the motor level that determines the choice. This conceptual change in the temporal hierarchy of processes also comes with a re-attribution of metacognitive noise: while most current models of confidence add noise to capture the reduced sensitivity of confidence compared to that predicted by decision sensitivity[1,49,50], here (at least some of) the additional noise can be attributed to confidence being computed prior to noise being filtered out by motor leakage. While the framework allows for ongoing primary evidence accumulation after decision commitment to contribute to post-decision confidence, further investigation is required to understand how other factors (for example, motor processes[51]) might additionally be incorporated into post-decision confidence.

While the experimental evidence presented here supports the central role of online confidence estimates in controlling behavioural efficiency, the evidence is inconclusive as to the mechanistic relationship between online confidence estimates and explicit metacognitive confidence reports (evaluated post-decision). Online confidence estimates used during deliberation could rely on representations of certainty from low-level (e.g., sensory) probabilistic (distributed) neural population codes[52]. Post-decision metacognitive confidence reports have been associated with a read-out of this low-level distributed coding in high-level brain regions[53] (frontal cortex). While our model formalisation of online confidence is consistent with this high-level read-out (conditioning certainty on the provisional choice), and we find the model online estimates predict post-decision confidence reports, the read-out for online confidence could rely on a distinct mechanism to that of explicit post-decisional confidence reports. We also propose that online confidence is used in decision-making irrespective of whether an observer expects to give an explicit report of post-decision confidence, and independently of how post-decision confidence is reported. These reporting factors could influence the read-out mechanism for post-decision confidence (for example, criterion noise[42]), without affecting online confidence (though this remains to be tested). Post-decision confidence has also been shown to incorporate additional information and biases, such as information accumulated after decision commitment[8,37,38], information about deliberation time[54] and motor preparation[51], biases toward confirmatory evidence[55], biases based on stimulus visibility[56] and attentional allocation[57], and developing detailed computational models of how these factors influence metacognition represents a long-term goal for the field[58].

In summary, in this work, we offer evidence that achieving behavioural efficiency during sensorimotor decisions involves the online coordination of perceptual, inference, motor, and metacognitive processes. These processes are typically studied as separate fields of research, with the limited examination of how they interact as a system. Here, we developed a computational framework to capture the coordinated effort of these processes and disentangle their underlying neural mechanisms. Importantly, this framework forms a new benchmark against which to continue to interrogate the neural systems involved in dynamic decision-making as well as characterise how these systems differ in neurodegenerative and neuropsychiatric disorders[59,60] and how they change across development[61] and aging[62].

## Methods
### Participants
Participants were recruited from the Experimental Subject Pool in the School of Psychology (University of Glasgow), who indicated the absence of diagnosed mental disorders and had normal or corrected-to-normal vision. All participants provided written, informed consent, and were reimbursed for their time at a rate of £6 per hour. We recruited 25 participants to meet our pre-registered plan (https://osf.io/5d8nh/[63]) to include 20 participants in the analysis, replacing participants whose performance did not rise significantly above chance

(58.3% correct, given 100 trials per condition, 2 excluded), whose reaction times were too slow (median > 2 s, the maximum duration of the stimulus, none excluded), or with poor quality EEG (3 excluded, for technical issues). Ethical approval for this study was granted by the College of Science and Engineering Ethics Committee at the University of Glasgow (application number 200210194). Data[64] and code[65] are made available of OSF.

### Materials
Stimuli were presented on a 68 cm ASUS G-SYNC™ monitor (2560 × 1440 pix, running at 120 Hz), controlled using Matlab and the Psychophysics toolbox[66–68]. Responses were entered via a Cedrus button box (RB-740, Cedrus Corporation). Gaze and pupil dilation were monitored using video-based pupil and corneal reflection (Tobii Pro X3-120 eye-tracker, sampling pupil size at 40 Hz), controlled using the Titta toolbox for Matlab[69]. EEG was recorded using a 64-channel BrainCap (EasyCap, GmbH).

### Stimuli
On each trial, observers were presented with up to 2 s of dynamic stimuli, formed of an array of 100 white dots whose position was updated at each frame, a variation of the classic random dot motion stimulus[70]. The classic stimulus was modified such that the previous dot position remained visible for 58 ms, appearing as if the dots trace outlines on the screen, meaning there is no ambiguity about the correspondence of one dot's position from frame to frame (but adding an explicit orientation cue to motion direction). Dot positions were updated in each frame according to a direction of motion (with speed 4 deg/sec) sampled from a circular Gaussian (Von Mises) distribution[71]. The direction was assigned to each dot for that dot's lifetime (83 ms), after which the dot was replaced. Dot lifetime was staggered such that in each frame, 10% of the dots were replaced by dots in new positions, with directions sampled from a new distribution. The mean and variance of the sampling distributions were updated throughout the first second of stimulus presentation to form nine conditions combining increasing, decreasing, or stable mean (angular distance from vertical) with increasing, decreasing, or stable variance. Dots sampled from a distribution with greater mean and low variance provide stronger evidence for the decision, whereas a smaller mean and high variance provide weaker evidence for the decision. The dots moved within a circular annulus with outer radius 4 degrees of visual angle (dva) and inner radius 1 dva. Dots whose positions moved outside the outer radius were replaced by dots with the same direction elsewhere. Dots whose positions moved inside the inner radius were transported to the other side of the inner annulus. A red fixation mark (diameter 0.12 dva) was presented at the centre and was present throughout each block of trials. The presented stimuli for all observers were sampled (with replacement) from a set of 360 unique stimuli with pre-defined dot directions.

The stimulus was removed immediately following a response (or after 2 s), and 200 ms later, the observer was cued to enter a confidence rating using a dial[53]. The dial was formed of the white outline of a circular annulus (radius 4 dva) with radial modulation of the annulus thickness to symbolise high (thick) and low (thin) confidence, with a line halfway to mark average confidence. The participant moved a white marker around the annulus by holding a response button until the marker reached their desired confidence. The marker moved at 1 cycle per second, and always started at low confidence, with unlimited revolutions to get the desired confidence report. The confidence dial was presented at a random orientation in each trial. This confidence dial design limited motor and eye-movement artefacts that could disrupt the EEG measures.

### Task
Participants sat in an adjustable chair, positioned such that their head was ~76 cm from the stimulus presentation monitor. On each trial, the

participant was asked to decide, as quickly and accurately as possible, if the dots were moving more leftward rightward or vertical, and then report their confidence that they made a correct decision. The stimulus continued (for up to two seconds) until the observer entered their response by pressing a button on the response box with their right hand. The confidence response was cued 200 ms after this and was followed by a 500 ms inter-trial interval. To prevent motion adaptation[72], in every second trial, the decision axis was vertically upwards and otherwise vertically downwards (where stimuli relative to the vertically downward decision axis appeared similar to falling snow, while vertically upward stimuli were rotated 180 degrees). Though the change in axis was entirely predictable, it was also cued each trial at fixation: the relevant arm of the fixation cross changed to black 300 ms before stimulus onset. Before beginning the experiment, the participant was shown four clear examples (leftward and rightward motion from vertically upwards and downward), and were given practice on 16 easy trials and 20 trials like those presented in the experiment, including practice with the confidence dial.

Participants performed a total of 900 trials (100 in each condition), for under 1 h duration. One participant performed 879 trials before a technical error (EEG battery depletion) prevented continuation, but was still included in the analysis. Participants were offered a break every 90-trial block. Before continuing after this break, participants were reminded to respond faster if their median reaction time for that block was greater than 1.5 seconds or to try to be more accurate if their average performance in that block was less than 60% correct.

## Analysis
**Behaviour.** Summary statistics (proportion correct, median response time, and median confidence) were examined to assess whether exclusion criteria were met, and differences across conditions were assessed using repeated measures ANOVAs with the appropriate transformations (d′, median log reaction time, and median z-scored confidence), post-hoc $t$ tests comparing pairs of conditions are presented in Supplementary Tables S1–S3. In addition, we assessed the relationship between discrimination performance, response times, and confidence, presented in Supplementary Fig. S1.

**Computational modelling.** The stimulus design enabled tracking of the objective decision evidence presented to the observer on each frame of each trial. The objective decision evidence was computed by estimating the mean and variance of the presented dot directions. The probability density of the Von Mises distribution at an angle $\theta$ is

$$V(\theta|\mu_t, \kappa_t) = \frac{e^{\kappa_t \cos(\theta - \mu_t)}}{2\pi I_0(\kappa_t)} \tag{1}$$

with $\mu_t$ and $\kappa_t$ the mean and concentration estimated from the presented dot directions at time $t$, and $I_0$, the modified Bessel function of order 0. The probability density to the left, $p(\mu_\theta < \theta_0|\mu_t, \kappa_t)$ is estimated by numerical integration,

$$p(\mu_\theta < \theta_0 | \mu_t, \kappa_t) = \frac{\int_{\theta_0 - \pi}^{\theta_0} V(\theta | \mu_t, \kappa_t) \mathrm{d}\theta}{\int_{\theta_0 - \pi}^{\theta_0 + \pi} V(\theta | \mu_t, \kappa_t) \mathrm{d}\theta} \tag{2}$$

where $\theta_0$ is vertical ($\frac{\pi}{2}$ for downward and $\frac{3\pi}{2}$ for upward). The evidence for a leftward response is $\nu_{L,t}$, the log probability that the dot direction was more leftward,

$$\nu_{L,t} = \log(\mathrm{p}(\mu_\theta < \theta_0 | \mu_t, \kappa_t)) \tag{3}$$

All models were constrained to this evidence as the basis of accumulation, as opposed to estimating drift rates. We validated this

assumption that this objective evidence informs behaviour using a GLM analysis to predict decisions and (separately) confidence from the objective evidence, which provided positive and mostly very strong coefficients for all observers (Supplementary Fig. S1).

**Classic single-integration model.** We implemented a version of a classic race/diffusion model of bounded evidence accumulation, in which there is no online control of decision processes that could respond to the dynamic changes in sensory evidence quality. The model accumulated evidence for left and right choices separately, determining the choice by the first to reach the bound (more akin to a race model). This was to make the framework more directly comparable to the double-integration model (below), but note that this formulation is equivalent to taking the difference in evidence to determine the choice based on an upper vs lower bound (more akin to a DDM). Accumulating the difference in log probabilities over time is a sequential probability ratio test, the Bayes optimal solution (in the case of no additive noise affecting the decision evidence).

The model has two free parameters describing decision processes ($\sigma$, the standard deviation of additive Gaussian noise, $\varepsilon \sim N(0, \sigma^2)$; and $b$, the height of the bound), and two free parameters describing non-decision time (for computational speed, non-decision time was sampled from the absolute of a Gaussian, $|N(\mu_U, \sigma_U^2)|$, which forms a right-skewed distribution similar to reaction times, so long as $\mu_U$ is small). Evidence accumulation takes the classic form,

$$x_{L,t} = x_{L,t-1} + \nu_{L,t} + \varepsilon_{L,t} \tag{4}$$

where $\nu_{L,t}$ is the current sample of evidence for a leftward choice (log probability based on the presented dots), and $\varepsilon_{L,t}$ is noise, independent from the previous sample and the sample applied to the rightward accumulator. Because of the stimulus evidence constraint, the timing of integration is constrained by the monitor refresh rate, $\Delta t = 0.0083s$. A leftward choice is made at time $t$, if:

$$x_{L,t} \geq b \cap x_{R,t} < b \tag{5}$$

(And vice-versa for a rightward choice). We also fit a model with non-linear bounds, requiring three free parameters describing a cumulative Weibull function[24], but the additional parameters were not parsimonious (Supplementary Fig. S2).

**Double-integration model with confidence control.** The classic model is the best process an observer can use, in the case that they have no knowledge of their own uncertainty. However, if the observer has knowledge of their own uncertainty, they could use this to optimise their decision processes further. Given that the noise affecting the evidence representation is temporally independent, the observer can use a moving average to actively filter out their own noise. They do this using a double-integration process, based on an extension of the Leaking Integrating Threshold model[19] (LIT).

The double-integration model has two accumulation stages. The first (primary, $x$) is the same as the classic model (Eq. 4). The secondary (motor, $y$) accumulator(s) re-integrate the already integrated evidence from the primary accumulator(s) with a smoothing factor, $\alpha$, determining the choice at a bound, $b$ (leaving the primary accumulator unbounded). We propose this smoothing is implemented mechanistically as a leakage in the motor accumulator. The smoothing is applied computationally using an exponentially weighted moving average:

$$y_{L,t} = y_{L,t-1} + \alpha_{L,t}(x_{L,t} - y_{L,t-1}) \tag{6}$$

This temporal smoothing increases the signal-to-noise ratio of the motor accumulator evidence: decreasing $\alpha$ increases the smoothing

but also slows the rate at which the evidence reaches the motor bound. The LIT used a value of $\alpha$ that is stable over time within a trial (but can be modulated across speed-accuracy trade-off conditions). The fit of this non-dynamic version is shown in Supplementary Fig. S2.

Here, we extended the LIT to incorporate confidence control, thus allowing dynamic modulation of decision processes according to fluctuations in sensory evidence quality. Although the general modelling approach was pre-registered, the exact implementation of confidence control was exploratory. First, the $\alpha$ parameter was set to be modulated by confidence in the choice, based on the primary accumulated evidence. The standard definition of confidence is the probability of the choice being correct[2,73]. For the ideal observer with no noise, confidence, $c$, equates to

$$c_{L,t} = \frac{p(L|x_{L,t})}{p(R|x_{R,t})}$$
$$= e^{x_{L,t} - x_{R,t}}$$
(7)

since the accumulated primary evidence is the sum of log probabilities. The human observer is affected by additional noise and uses an estimate of the variability of this noise, $\sigma_c^2$, to weight their confidence according to a sigmoid function:

$$c_{L,t} = \frac{1}{1 + e^{-\left(\frac{x_{L,t} - x_{R,t}}{\sigma_c^2}\right)}}$$
(8)

$$\frac{1}{\sigma_c^2} = \frac{\beta}{\beta + \sigma_t^2}$$
(9)

The additional parameter, $\beta$, modulates the relative weight of the cumulative variability of the decision evidence, $\sigma_t^2$, which is assumed to be monitored online. This can be interpreted as over- or under-estimating the variability itself and/or the probability ratio of the evidence for each choice. The $\alpha$ parameter controlling the temporal smoothing factor is then set (for each choice separately):

$$\alpha_t = (c_t - p_c) \times (c_t > p_c)$$
(10)

With $p_c = 0.5$, the motor accumulator only integrates evidence for choices when confidence in a correct choice is greater than chance.

The effect of the external evidence variability on choices and response times (which slows down decisions to a greater extent than manipulating the mean; Fig. 1c) can be captured via a multiplicative effect on internal noise, such that $\varepsilon \sim N(0, \sigma_t^2)$, with $\sigma_t^2$ equal to the internal variability parameter multiplied by the circular variance of the external stimulus ($1 - \frac{I_1(\kappa_t)}{I_0(\kappa_t)}$).

Other models employing an online updating of decision commitment strategies[6,13,16], incorporate some form of cost function to minimise the time and effort of decision processes. We tested a simple form of cost function by assuming the observer aims to commit to their decision by stimulus offset (2 s from stimulus onset). We implemented this at the level of the temporal smoothing parameter by assuming a linear decrease in $p_c$ from some time, $t_d$, such that $p_c = 0$ by stimulus offset. In this way, the observer can implement less temporal smoothing with lower confidence, pushing lower-quality evidence toward the motor bound as the urgency to commit to a decision increases.

Finally, we found we were able to simplify the model by fixing the bound parameter across participants. Thus, the final model had five free parameters, three describing decision processes via the internal noise ($\sigma$), the relative weight of this noise for confidence ($\beta$) and the time constant of the cost-function ($t_d$), plus the two parameters for non-decision time. Supplementary Fig. S2 shows the effects of modulating these parameters on behaviour.

As an additional check, we examined whether a weighting on the variability of the decision evidence could explain behaviour in a bottom-up manner, that is, affecting the primary evidence as it is accumulated in a single-integration framework (Fig. 3b). We implemented this by applying the weighting described by Eq. 9 on the primary evidence as it is accumulated.

$$x_{L,t} = x_{L,t-1} + \frac{\beta}{\beta + \sigma_t^2} \nu_{L,t} + \varepsilon_{L,t}$$
(11)

This formalisation proved worse at the computational level, as well as providing a poorer prediction of the EEG, and cannot explain post-decision confidence without further modifications (Supplementary Fig. S6). While bottom-up explanations may be appealing in terms of their mechanistic simplicity, the double-integration framework could be considered more comprehensive in terms of the additional behaviour and neural processes it encompasses.

**Model fitting and comparison.** Parameters were fit to minimise the negative log-likelihood (NLL) of choices and response times (RT) using Bayesian Adaptive Direct Search[74]. A Monte Carlo simulation approach was used to estimate the NLL over 10 quantiles of choice-RT data[75], with 2000 simulations for each stimulus. Models were compared using the Bayesian Information Criterion (BIC) and exceedance probabilities[76,77].

**Pupillometry.** Pupil data were pre-processed by interpolating outlier samples and then using a moving average filter (window size 4 samples/160 ms). Blinks were identified and interpolated (5 sample windows) based on 10x the median absolute deviation in the derivative. Further outlier samples were identified by iteratively interpolating over samples that differed from the trend by more than 10x the median absolute deviation until no additional outliers were identified (or up to 10 iterations). This differed from our pre-registered plan to use a standard deviation criterion, but the median absolute deviation is more in line with established recommendations[78]. Pre-processing was performed on the data from each eye separately before averaging the eyes and then z-scoring. Data were then epoched into two time windows of interest: from −0.5 to 2 s around the time of stimulus onset; and from −2 to 1 s around the time of the response. Trials where a time window contained more than 30% of samples interpolated were removed from further analysis. Two participants' data were removed from further analysis because more than 1/3 of the trials were removed in this process. Data were baselined to the average from 500 ms prior to stimulus onset. We performed an exploratory analysis, presented in Supplementary Fig. S5.

**EEG.** EEG data were pre-processed using the PREP pipeline[79] implemented in EEGlab[80]: line noise was removed (notch filter at 50 Hz), and robust average re-referencing was applied on detrended data (1 Hz). Data were then filtered between 0.5 and 80 Hz and downsampled to 250 Hz. Independent Components Analysis was used to remove artefacts caused by blinks and excessive muscle movement.

A preliminary, classical EEG analysis was conducted: Central Parietal Positivity (CPP[43]; which frequently aligns with decoded evidence accumulation[81]) and Response Readiness Potentials (RRP[82]) were extracted from the data epoched around the time of the response (from 2 s prior to 1 s after the response). These classical EEG response functions were compared across behavioural variables, as reported in Supplementary Fig. S3. This analysis may be more directly comparable to previous work. The classic EEG components show typical dynamics, but also align with several predictions of the double-integration framework: CPP ramps up to the time of the response, but doesn't appear to terminate at a single bound; RRP also appears to ramp prior to the response, with differences between fast and slow responses leading up

to the response, but only small differences by the time of the response itself (consistent with a bounded accumulator). This offers indirect support for the proposed role of associative and motor processes under the double-integration framework.

**Decoding analysis.** The decoding analysis of the EEG used an inverted linear encoding model to estimate weights on sensor activity to best predict variables of interest.

The inverted linear encoding model[83,84], assumes the EEG sensor signals ($Y$) encode a linear weighting ($w$) of the variable of interest ($X$), estimated by ordinary least-squares:

$$\widehat{w} = \left(X^T X\right)^{-1} X^T Y \qquad (12)$$

The variable of interest is then predicted by applying these weights to held-out data (10-fold cross-validation):

$$\widehat{X} = \left(\widehat{w}^T \widehat{w}\right)^{-1} \widehat{w}^T Y \qquad (13)$$

We examined how the model-predicted (primary and motor) evidence accumulation might be represented in the EEG sensor signals using this decoding analysis. To do this, we simulated evidence accumulation in each trial, using the double-integration framework and the parameters fit to each participant. For each trial, we took the median of the simulated evidence accumulation traces that predicted the observer's behaviour (same choice made within 100 ms of their response time). Both the simulated primary and motor accumulators (for the chosen option) tended to increase over time and were strongly correlated over time (average Fisher transformed correlation $z = 1.02$). This correlation was broken by taking the derivative, where the slope of the primary accumulator in the lead-up to the response is much flatter than that of the motor accumulator ($z = 0.23$). To match the sampling frequency of the evidence (120 Hz) with the sampling frequency of the EEG (250 Hz), the slopes of each EEG sensor were calculated over a 6-sample window (linear regression coefficient; ~42 Hz), while the slope of the simulated evidence was calculated over 3 samples (40 Hz). We found that applying a lowpass filter (8 Hz) on the EEG data prior to this stage improved the reliability of the decoding analysis (a comparison of filter choices is shown in Supplementary Fig. S4). We then applied the above decoding analysis to estimate the weights ($w$) on EEG sensors ($Y$) that predict the simulated evidence ($X$) in the 400 ms window prior to the response (a comparison of windows of 300 and 500 ms is shown in Supplementary Fig. S4). To maximally distinguish EEG signals corresponding to the primary and motor evidence, weights were estimated simultaneously (with an intercept term, $X$ and $w$ are of size n x 3).

The precision of the decoding can then be assessed by taking the correlation between the EEG-predicted evidence and the model simulated evidence (applying a Fisher transform for normalisation, for the primary accumulator, mean $z = 0.024$, t(18) = 5.90, $p = 1.10 \times 10^{-5}$; for the motor accumulator, mean $z = 0.086$, t(18) = 9.07, $p = 2.47 \times 10^{-8}$). The precision computed separately at each sensor gives an indication of the topography of the contributions to the EEG representation (shown in Fig. 4b).

The same approach was used to decode post-decision confidence, but in this case, each trial is associated with a single value of post-decision confidence (from the behavioural ratings) that is assumed to emerge from neural processes around the time of the response. For this reason, we examined decoding precision using EEG data at each time point in the response-locked epochs separately. We found precision tended to increase in the lead-up to the response and continue increasing after, however, there was a dip in decoding precision at 200 ms following the response (presumably due to interference of the confidence response cue). We therefore took the weights estimated

from the time-point with the greatest precision for each participant from the time window 0–200 ms following the response. We used these weights to estimate online confidence by applying them over time in the lead-up to the response.

**Granger causal analysis.** For each participant, we calculated the EEG-predicted motor and confidence components by applying the weights estimated using the decoding analysis above to the full (not epoched) timeseries. Large windows of non-task-related recordings (greater than twice the maximum tested lag of 800 ms; for example, breaks between blocks) were excluded. For each participant, vector autoregressive models were estimated with lags from 200 to 800 ms (50 to 200 samples) and the model with the minimum Akaike Information Criterion was selected (on average, lags were 570 ms, ranging from 272 to 784 ms). All models were found to be stable. Leave-one-out Granger causal tests were conducted within subjects, and all participants showed significant evidence against the null hypothesis to exclude the history of the confidence component in predicting the motor component (in addition to the motor component itself).

## Reporting summary
Further information on research design is available in the Nature Portfolio Reporting Summary linked to this article.

## Data availability
The data used in this study (raw behaviour in CSV and Matlab format, raw EEG, and raw pupillometry) are available on the Open Science Framework: https://doi.org/10.17605/OSF.IO/5D8NH.

## Code availability
Analysis code scripts are available on the Open Science Framework: https://doi.org/10.17605/OSF.IO/5D8NH.

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

## Acknowledgements

This work was supported by a European Research Council consolidator grant (865003) to M.G.P.

## Author contributions

Conceptualisation: T.B. and M.G.P.; Methodology: T.B. and M.G.P.; Software: T.B. and M.G.P.; Formal Analysis: T.B.; Investigation: T.B.; Resources: M.G.P.; Data Curation: T.B.; Writing – Original Draft: T.B.; Writing – Review & Editing: T.B. and M.G.P.; Visualisation: T.B. and M.G.P.; Supervision: M.G.P.; Project Administration: M.G.P.; Funding Acquisition: M.G.P.

## Competing interests

The authors declare no competing interests.
