## [Peer Review File · Nature Communications]

REVIEWER COMMENTS

Reviewer #1 (Remarks to the Author):

The manuscript titled "Confidence control drives behavioural efficiency in dynamic environments" by Baldstrom and Philiastides analyzes behavioral and EEG data to test different models of confidence construction. Through a clever manipulation of stimuli - which change over time in some conditions - they manage to discern which type of model best accounts for the data (a model proposed by the authors in previous work, based on two accumulators instead of the more traditional single accumulator). Furthermore, EEG and pupillometry data support these findings both in decoding analyses and in more traditional ERPs analyses. Authors find the work very creative, well-done, and highly relevant to the field. We have some comments which we'll state below:

- The confidence response is unusual, very uncommon. Why did the authors decide that confidence should be reported on a continuous bar? Can participants move backward on this bar? Furthermore, trials with high confidence will have much higher confidence response times and therefore longer deliberation times, exactly correlated, than low confidence trials, as there is a bar to traverse. What is the rationale?
- Figure 1b is unclear, particularly in identifying conditions based on colors (a code that is maintained throughout the work). We suggest clearer color coding of conditions, perhaps with a small table.
- In Figure 3, it is shown that double accumulation models fit better than single accumulation ones, but there is no numerical comparison to quantify this claim, only in the supplementary. We suggest including this in the main text.
- The hypothesis of a double accumulator as a motor accumulator is presented in the introduction without much further justification. While there is a previous paper that accounts for this, it would be good for the introduction to provide better justification for the proposed analogy.
- The pupillometry results are much weaker and more preliminary than the rest, only supported by a slight correlation between the filtered EEG signal and pupil size, which would not withstand multiple comparisons if other frequencies were included. We suggest moving them to supplementary material.

- There is no parameter recovery analysis. While the authors make no claims about differences in behavior captured by the parameters (though they do show some of this in Figure 2 of the supplementary material), it would be good to see this, especially if you intend the model to be somewhat generalizable and applicable. In Figure 2 of the supplementary material, it would be good if you clarify whether you fixed the other parameter values when varying one in particular, and if so, to what values you did so.
- Along the same lines as above, we are curious about the beta parameter that reflects the weight of evidence variability in confidence control (controlling over/under confidence). Are participants with better metacognitive sensitivity (Fig 1E) more calibrated with respect to this parameter??? It could be an extra way to validate the generative process of the model.
- The red line in Figure 1D is the average confidence in errors, whereas everything else is the median confidence. It should be labeled as the median confidence in errors.
- You perform t-tests between all conditions to show differences. We believe a proper post hoc test after ANOVA to correct for the number of tests conducted would be better.
- In the results of the tests, the F statistic is larger in confidence when the a priori effect seems smaller.
- The way average confidence is reported seems odd; subjects cannot go back on their report.
- In line 338 of the discussion, I think it's poorly worded, could it be? It should be something like "at the motor bound" instead of "that the motor bound"??
- We didn't understand what you mean in the methods in the 'task' part with 'To prevent motion adaptation, every second trial the decision axis was vertically upwards and otherwise vertically downwards' (lines 421-422).
- In the EEG methods, you sometimes mention a frequency of 256 and in other lines, you mention 250 Hz. Overall, the methods are too concise regarding EEG data preprocessing; the authors mention an inverted decoding model without much further specification.

Reviewer #2 (Remarks to the Author):

This is a behavioral & EEG study which explores the role of confidence in online control of the decision process. The question is important because agents need to adaptively regulate the integration of evidence to perform optimally in dynamic environments. The model is quite elegant, especially the distinction between primary and motor accumulators which could resolve a number of debates and contradictory findings in the field.

However, I had some concerns about the model comparison and broader claims of superiority over previous models. I also found the neural evidence not very convincing, and at times felt like the paper would be better without it, even if this weakens the overall impact. But I am open to counter-arguments, and hope that the comments are found to be useful for improving the manuscript.

Major comment (1):

It is reasonable to claim that the classic model is “unable to describe both how participants could be faster and more accurate with early high-quality evidence while also slowing down with early low-quality evidence.” But I don’t understand how it cannot even fit the red and lavender conditions, where evidence strength is stable. More generally, what constrains the classic model to such a tiny range of RT (0.93-0.99s) rather than spanning the full empirical range, which after all is bracketed by conditions that are not the critical conditions of the experiment?

I assume it’s because the evidence-varying conditions are interleaved with the stable ones, and perhaps the fit shown in 3A is quantitatively better than a hypothetical one which spanned the full RT range but missed the middle data points. (Or maybe there were there fewer trials of the stable conditions, so those would have less leverage on the fit?) But if so, this intuition is not spelled out, so it’s easy to be skeptical that alternative models were given a fair shake.

Major comment (2):

The key neural result is the correlation between corresponding traces in Fig 4A, bottom left, and Fig. 4B, left. But other than the lavender trace being above the others for the latter two-thirds of the

graph, I don't really see a correlation here. And in the first third, the ordinal relationship is largely reversed across the two plots.

Perhaps there could be a better way to display this finding? One could try reducing the number of traces (e.g. collapsing to 4 conditions, as suggested above), or showing the pairwise time series data that make up the correlations, using a scatter plot. But as it stands I'm not sure the result is compelling enough to support a claim about neural mechanism, notwithstanding the interesting anatomical segregation of motor vs. central-parietal.

Major comment (3):

207-214: It's impressive that the final state of the model-simulated online confidence predicted observers confidence reports, but I don't think it's fair to say that the current model offers a more natural explanation for confidence ratings compared to previous work. The current model gives an expression (Eq 8+9) for confidence but no plausible mechanism for reading it out, whereas previous studies offer a variety of mechanistic or process models (perhaps heuristic, yet often approximately optimal) for post-decision confidence.

Also, "since the bound on the primary accumulator implies the same quantity of evidence for all choices" seems incorrect to me. In a 2D accumulator, as in Kiani et al. 2014 and here, the state of the losing accumulator varies across trials, providing an explanation for variability in confidence (incorporating elapsed time) despite a fixed value of accumulated evidence for the winner.

Major comment (4):

300-317: This sounds like a cool idea but it comes a bit out of nowhere, and I failed to see how the analysis adds to the story about confidence control. If pupil dilation reflects commitment, faster dilation would be associated with faster responses regardless of confidence control. The link with beta power is also not well developed, and empirically weak (pupil data are too variable to trust the bottom-right of 4F). Nothing wrong with exploratory analyses, but I'm not sure this one logically supports the claim being made, even in principle.

Minor comments:

57-60: This part nicely situates the work relative to Zylberberg et al. and Levi et al., though I might suggest adding a nod to Drugowitsch, Moreno-Bote, and Pouget (NeurIPS 2014, Optimal decision-making with time-varying evidence reliability) if the authors agree it is relevant.

102: Behavioural performance?

140: At this point I was reminded of the distinction made by Pouget et al. (Nat Nsci 2016) between confidence and (un)certainty. The former is conditioned on a choice ($P(\text{correct} | \text{choice, evidence})$) whereas the latter is basically the inverse variance (or entropy) of the posterior over the stimulus dimension (or category). It sounds like the current study wants to claim there is a provisional degree of confidence calculated at every time point (Eq. 7), but then this is modulated by the variability of an additional noise term (Eq 8) incorporating a weighted estimate of the cumulative variability of the evidence — which sounds more like uncertainty, by the Pouget definition. (See also Li et al., Optimal policy for uncertainty estimation concurrent with decision making, Cell Reports 2023.)

Of course the authors are welcome to call it whatever they want, but I think it matters whether the data actually support the idea that the brain continuously assigns a degree of confidence to a provisional choice at each moment, versus merely estimating the variability of the evidence (+ noise), as these two accounts may point to distinct neural mechanisms and make different predictions for subsequent work. Any clarity on this topic would I think be useful to add, perhaps in the Discussion.

Fig 3: Having left Fig 1 behind I found it difficult to remember which color corresponds to which condition. It might help to reinsert Fig 1B here (and/or in Fig 4) as an inset. But a more radical idea is to regroup the conditions as, say, 'stable-high, stable-low, early-high and early-low,' since no key point is being made about changing the mean vs. variance. This would make all the graphs less spaghetti-like, including Fig. 4A+B and Supp Fig S2 which are very hard to parse. Just a suggestion.

Fig. 3E: Why not show the unity slope lines, as in D?

241: What is the interpretation of the derivative of an accumulator? Is this just a way to maximize the differentiation between primary and motor, or is it because of the need to compare with the derivatives of EEG signals?

248: Speaking of which, what aspect of the signal is being extracted? Just the raw signal, or power in some frequency band? This relates to the above point that we are not given intuition for why the derivative is used.

249: Why is this called precision?

Fig 4C,D: Not everyone is familiar with these EEG array plots; could you add a contour/boundary to illustrate the location of motor and parietal cortices (determined independently from the activity heat map)? Also, there's no legend for what red vs blue means.

293: Only the last 200 ms are used, understandably, but confidence control (in this task) is required from trial onset. Is the assumption that the strength of confidence control fluctuates across but not within trials, such that estimating it using the final 200 ms is sufficient to apply that to the whole trial?

310, 314: Oddly coincidental that both of these p values are exactly 0.018. And is it valid to make a conclusion of significance based on the mean of multiple p values?

392: direct -> direction

426: Were Ss informed about the dynamics of stimulus strength? Or if you asked them afterwards, did they report that they detected that something was changing in the first second? I think it's worth noting/speculating whether the predictable timing of the transition at $t=1s$ had anything to do with the clustering of RTs around that time (most between 0.85-1.15s, doesn't seem like an accident).

There are two Supp Fig S5's (the 'bottom-up' one should be S6 I guess)

Reviewer #3 (Remarks to the Author):

In this paper, Balsdon and Philiastides test the performance of a two-accumulator model of perceptual evidence in its ability to explain participants' decisions to respond, or to wait and accumulate further evidence before committing to a decision. They also seek neural correlates of these two proposed accumulators (finding that they are plausibly separate, in neural terms, and that the second may be associated with motor signals).

This is a fantastic study: interesting, timely, and creative in its experimental design. The analyses are sound, carefully conducted, and clearly explained. I congratulate the authors for their good

work, that I thoroughly enjoyed reading and thinking about. This study is computationally very close to the own authors previous work, so I think some further clarification would help explain the conceptual novelty of their findings. I have, additionally, a few comments or questions.

Major comments:

- Interpretation of Figure 1E: The authors hypothesised that participants with a tighter relationship between confidence and decision accuracy would also show a stronger modulation of the experimental manipulations. This is fine, but I can think of a few possible confounds for why this effect might emerge.

i. First, if I understood it correctly, the results of the GLM they used to measure the association between confidence and performance depends on the within-subject range of performances. I.e, a participant with a narrower performance range will have a less clear association with confidence. So it's not a given that this measure can be used to compare different participants.

ii. Also, (and I hate to be 'that guy', but it should be said) simple and general motivational or attentional effects could explain this relationship. A participant that is more attentive throughout the task will likely show a stronger effect of the manipulation and also have a better association with confidence. Attentional effects are often a trivial (and admittedly boring) suggestion as alternative interpretations, but the authors could perhaps make it clear that the effect they see is compatible with their hypothesis, but does not provide direct evidence for it, unless they can show in a control analysis that this effect is not general.

- In line with my previous comment, I am not sure that the authors' interpretation of the signal c as 'confidence' is granted, and it is definitely not shown. If I missed something, I apologise. The model poses that a readout from the first accumulator c is used to modulate the second accumulator. But the authors never test, for example, that the modelled final value of c corresponds to participants' reports of confidence. Indeed, I am a little bit surprised that the paper is so centred on the role of online confidence on decisions (even having it on the title) but there is so little emphasis on the analysis of participants' actual confidence reports. I couldn't access the preregistration (it's embargoed), but it would be good to know if the authors had planned any more analyses of participants' explicit confidence reports.

- I am not an expert in evidence accumulation models, so I cannot judge how novel this model is. However, the authors already presented and tested this double accumulator model (e.g. ref 16). Here, the argument is that this model not only better predicts decisions in different experimentally-controlled conditions of speed-accuracy tradeoffs, but also explains participants' own cognitive control. That is fine, but is by no means a novel argument, as previous studies have shown that confidence drives decisions to seek further information (e.g. Desender Psych Sci 2018). In fact, these other studies have shown it less ambiguously, as it was confidence reports themselves, and not a latent variable that is assumed to correspond to confidence. Perhaps the authors would argue

that the novel aspect is the volatility of the environment, and how much better the two-accumulator model fares as compared to a single-accumulator model. Either way, given how much (excellent) prior work the authors have done on this topic, it would be good, I think, to better explain in the discussion what new piece of the puzzle they have found in this specific study.

- Could the authors explain why the confidence was reported in the way it was? Could this have led to any kind of confound in that more motor preparation for a longer response correlated with higher confidence, hence appearing as a motor topography of the second accumulator? (I understand that this is 400 ms prior to the discrimination response, but also confidence is argued to be computed online, so perhaps the two are not separable).

- Related to the comment above, a more general/conceptual question: Do the authors speculate that this second (motor) accumulator would be implemented differently in neural terms, if the confidence response were provided differently? My reading of the paper is that, because confidence is used online for decisions to commit to a decision or wait, it would exist even if participants hadn't been asked to rate confidence at all. Is this what the authors argue? I think that if the authors really want to make such a strong claim about confidence, this might be a possible control experiment that could help their argument.

- The two-accumulator model is very different from recent accounts (admittedly in detection contexts) suggesting that a single accumulator model is enough to explain confidence as the difference between maximal evidence and the distance-to-bound. (Pereira et al, Nat Comms 2021). Perhaps the changing signal-to-noise ratio in the task shown here is enough to explain the difference, but I was surprised to see no reference at all to this work.

Minor comments:

- Line 19: I would suggest to remove the motivation to study this based on the (perhaps simplistic) assumption that of what would and not be inefficient for the brain to do.

- Figure 4.A: are the colours mapped wrong? Would one not expect the purple line to accumulate evidence slowest?

- Line 417: minor typo? "More to the leftward or rightward of vertical" seems grammatically wrong.

- Line 421: I did not understand the difference between vertically upwards or downwards. Was the decision not based on a full vertical midline of the stimulus?

Reviewer #4 (Remarks to the Author):

Reviewer #1 (Remarks to the Author):

The manuscript titled "Confidence control drives behavioural efficiency in dynamic environments" by Baldstrom and Philiastides analyzes behavioral and EEG data to test different models of confidence construction. Through a clever manipulation of stimuli - which change over time in some conditions - they manage to discern which type of model best accounts for the data (a model proposed by the authors in previous work, based on two accumulators instead of the more traditional single accumulator). Furthermore, EEG and pupillometry data support these findings both in decoding analyses and in more traditional ERPs analyses. Authors find the work very creative, well-done, and highly relevant to the field. We have some comments which we'll state below:

We thank the reviewers for their insightful and constructive comments, which have greatly helped us to improve the manuscript. To ensure we have fully addressed all comments, we have numbered each comment according to reviewer number and comment number, such that R1C2 corresponds to the second comment of this review. We provide a track-changed document with all changes commented with this code. In responding to reviewers, we quote the text, referring to the page and line number of the clean (not tracked-changed) revised manuscript, should the reviewer wish to find the changes in the clean manuscript also.

R1C1

- The confidence response is unusual, very uncommon. Why did the authors decide that confidence should be reported on a continuous bar? Can participants move backward on this bar? Furthermore, trials with high confidence will have much higher confidence response times and therefore longer deliberation times, exactly correlated, than low confidence trials, as there is a bar to traverse. What is the rationale?

We thank the reviewers for this comment. We based the confidence dial on Geurts et al., Nature Human Behaviour (2022) (now explicitly referenced in the methods). The circular confidence response stimulus doesn't contain systematic orientation information (where the perceptual decision is orientation based). We additionally prioritised minimising EEG artefacts that can arise from other confidence report methods: 1) discrete ratings (e.g. 1-4) require multiple buttons that can increase movement artefacts in the EEG; 2) a horizontal scale would mean eye-movements from looking at the scale would correlate with confidence (looking leftward for low confidence and rightward for high - creating an artefactual signal that can be used to decode confidence in the EEG; though it is unrelated to the neural processing for computing confidence per se).

We have added this rationale for the choice of confidence response method in the methods section P14L413:

“The participant moved a white marker around the annulus by holding a response button until the marker reached their desired confidence. The marker moved at 1 cycle per second, and always started at low, with unlimited revolutions to get the desired confidence report. The confidence dial was presented at a random orientation each trial. This confidence dial design limited motor and eye-movement artefacts that could disrupt the EEG measures.”

We do not think the confidence response dial constrained confidence deliberation time to be dependent on the confidence reported. Participants could deliberate before starting to enter their confidence response, and we see substantial variability in the time from the onset of the confidence dial to enter the confidence report (y-axis of the figure below) that is independent of the confidence report (x-axis of the figure below; though there is a lower bound dependent on the confidence report):

Each marker in the figure is one trial from one participant; all trials with confidence RT less than 2 s are shown.

To address this comment in the manuscript we have made it clear in the Figure legend (Figure 1a) that more than one revolution was possible (to a lower confidence level) and participants could initiate their confidence response with unlimited time P4L96:

“After 200 ms observers were cued to make a confidence rating, entered after unlimited time by holding a response key to move the circular marker around the annulus until it reached their desired confidence (thicker annulus means higher confidence; it was possible to make more than one revolution to return to a lower confidence).”

We agree that it is more common to ask participants for a rating on a discrete scale: Studies using a 4-point confidence rating make up over 50% of those registered on the confidence database (93/174; <https://osf.io/s46pr/>). The discrete rating is often more practical, it is easy to use and explain to participants, and as the reviewers point out, it allows for more direct measurement of confidence response times. We note that about 30% (52/174) of studies registered on the confidence database use a continuous scale. While confidence response times are interesting in their own right, they were not the focus of the current work, and we felt it more important to minimise the EEG artefacts.

R1C2

- Figure 1b is unclear, particularly in identifying conditions based on colors (a code that is maintained throughout the work). We suggest clearer color coding of conditions, perhaps with a small table.

We thank the reviewers for this suggestion which is of great importance for understanding the results presented in the figures. For the colours, it was quite difficult to find colours that are both distinguishable across the 9 conditions while informative of the condition category (we had already changed the colour coding several times during the preparation of the manuscript to get something useful).

We have instead made two additions to the manuscript.

First, we added a table as suggested, to Figure 1b (copied below).

Figure 1b:

“b) The quality of the sensory evidence was manipulated according to nine conditions created by adjusting the parameters of the circular Gaussian (von Mises) distribution from which the dot directions were sampled. The table to the left shows the parameter values in each condition: the mean difference from vertical (degrees) and the Kappa parameter which controls the inverse spread of the distribution (arrows highlight the change in terms of evidence strength, which is also depicted to the right). In four conditions the evidence was systematically increased or decreased over the first second of stimulus presentation (by increasing/decreasing the mean or the variance). In three conditions the evidence was completely stable. Two additional conditions of moderate evidence were created by changing the mean and variance in opposite directions.”

Second, as suggested by another reviewer (R2C8), we have reinserted the schematic from Fig 1b as an inset to remind the reader of the colour coding in future Figures. For example, in Figure 3a:

We hope these changes help address this comment.

R1C3

• In Figure 3, it is shown that double accumulation models fit better than single accumulation ones, but there is no numerical comparison to quantify this claim, only in the supplementary. We suggest including this in the main text.

We apologise this was not made clear enough in the previous manuscript. Formal model comparison statistics were provided in the main text in the paragraph under the results section titled “Computational evidence for online confidence control” (line 193 of the previous manuscript). To make these less concealed, we have rephrased this paragraph and separated out the numerical comparison to a separate paragraph (some additional changes have also been made in response to R2C1 and R3C3) P7L193:

“The classic single-integration model dramatically failed to capture the pattern of behaviour across intermixed stimulus conditions (**Figure 3A**). The model was unable to describe both how participants could be faster and more accurate with early high-quality evidence while also

slowing down with early low-quality evidence. This failure is in part due to the fact that the model has to implement a single decision bound across the intermixed stimulus conditions, and in part due to constraining all models to use the evidence from the presented stimuli (a constraint validated by a separate GLM analysis, **Supplementary Figure S1**). Similar model failures were also apparent for a variant of the single-integration model with time-varying bounds, and for a static LIT model (as implemented in previous work^{18,19}) where double-integration occurs with a smoothing parameter that is stable over time (see **Supplementary Figure S2**). Describing behaviour in this dynamic sensory environment requires a model where decision processes can react to the unpredictable transitions in the evidence quality online (**Supplementary Figure S6**), which is provided by the double-integration model with online confidence control (**Figure 3B**; and would otherwise require effectively stimulus-time-dependent bounds on the primary evidence^{13,14}, **Supplementary Figure S2**).

Formal model comparison suggested the double-integration model with online confidence control proved vastly superior to the classic model (**Figure 3C**; $\sum BIC_{classic} = 8.33 \times 10^4$, $\sum BIC_{confcontrol} = 8.25 \times 10^4$, $\sum \Delta BIC = 842.61$, protected exceedance probability = 0.981; see **Supplementary Table S4** for a summary of parameters and fit statistics).”

R1C4

- The hypothesis of a double accumulator as a motor accumulator is presented in the introduction without much further justification. While there is a previous paper that accounts for this, it would be good for the introduction to provide better justification for the proposed analogy.

We thank the reviewers for highlighting the lack of clarity in the introduction. In fact, the secondary integrator was inserted because of the rapidly emerging evidence for a more active role of motor processes in decision-making, including clear accumulation profiles at the level of (pre)motor structures. As opposed to suggesting motor processes justify secondary integration, secondary integration is proposed to capture the computational function of motor processes. We have made this clearer on P2L39:

“The inclusion of a secondary accumulator in the double-integration framework was motivated by its neurobiological plausibility: this secondary accumulation is proposed to take place at the level of motor processes, to explain the build-up of motor activity prior to decision commitment^{25,26}, and evidence for a more active role of these motor processes in decision-making^{27,28}.”

And when further describing the model in the results P5L114:

“The LIT model employs a double-integration framework which was initially motivated by neurobiological plausibility...”

R1C5

- The pupillometry results are much weaker and more preliminary than the rest, only supported by a slight correlation between the filtered EEG signal and pupil size, which would not withstand multiple comparisons if other frequencies were included. We suggest moving them to supplementary material.

We agree with the reviewers, and this sentiment was also shared by Reviewer 2 (R2C4). We have moved these results to the supplementary materials (Supplementary Figure S5, copied below). For openness, we include the description of obtaining pupil measurements in the methods and refer the reader to the supplementary section for this exploratory analysis.

“Supplementary Figure S5. Decision preparation signals. A) Effect of response speed (top row) and confidence (bottom row) on pupil size, pupil derivative, and motor beta power. In all cases a median split was used, within participants. Shaded error shows 95% within-subjects confidence (difference between split). Grey shaded regions show significant differences at the group level. Motor beta power was computed from the power spectrum across frequency tapers from 16 to 32 Hz with 25% spectral smoothing, resolved using wavelet convolution in FieldTrip. Channels were selected for each participant as the channel with the greatest (negative slope on beta power in the response epoch (C3 for 8 participants, and FC1, C1, and FC3 for 5, 4, and 3 participants respectively). B) Difference in the pupil derivative (velocity of change in pupil size; black) and beta power (red) using a median split of response times (fast – slow RTs). To make beta power better comparable with pupil size, the y-axis is reversed, such that decreased beta power corresponds to increased pupil size, the direction thought to indicate response preparation. Shaded regions show the 95% between-subjects confidence intervals. Thick lines at $y = 0$ mark regions of cluster-corrected significance for the difference in pupil velocity (black) and beta power (red) based on the median split of response times. Fast responses were associated with faster pupil dilation following stimulus onset (cluster corrected significant window, 0.125 to 1.15 s; mean $t(18) = 3.28$, mean $p = 0.018$). The same median split of response times showed an appreciably similar effect on motor Beta power (red line; cluster corrected significant window 0.28 to 0.79 s; mean $t(18) = 2.75$, mean $p = 0.018$; the effects in the response-locked epoch also follow a similar shape, but the pupil derivative is too variable to reach significance). While preliminary in nature, this finding is in line with the hypothesis that motor leakage is implemented via subcortical inhibition, reflected in part in pupil dilation and resulting in decreased cortical motor Beta power with the release from inhibition.”

RIC6

• There is no parameter recovery analysis. While the authors make no claims about differences in behavior captured by the parameters (though they do show some of this in Figure 2 of the supplementary material), it would be good to see this, especially if you intend the model to be somewhat generalizable and applicable. In Figure 2 of the supplementary material, it would be good if you clarify whether you fixed the other parameter values when varying one in particular, and if so, to what values you did so.

We thank the reviewers for pointing this out, an oversight on our part. We now include a parameter recovery analysis in Supplementary Figure S2. Following the suggestions of Wilson & Colins (2019) eLife, and given this parameter recovery takes some time (since there is no analytic solution for the model), we only performed the parameter recovery for the best fitting model, and constrained the sampling to the range of parameters fit to the human data. We now also state explicitly that in the simulation of Supplementary S2F (showing the effects of each parameter), the other parameters were fixed to the median, from Supplementary Table S4.

“**Supplementary Figure S2. Comparison of models and parameters. ... F)** Demonstration of the effects of three critical model parameters on behaviour: the weight on evidence variability (β ; left); the urgency parameter, marking time at which motor smoothing is influenced by lower confidence (t_d ; middle); and decision noise. The small, medium, and large (thin to thick) lines correspond to the minimum, median, and maximum value of the parameters fit to participants, the other parameters were fixed to the median (see Supplementary Table S4). Both proportion correct and response time are influenced by all three parameters. **G)** Parameter recovery analysis for the parameters of the best-fitting model. Parameter values were sampled from a uniform distribution around the range of values fit to participant data. Behaviour in the experiment (with the same number of trials) was simulated using the model, and then the model was fit to this simulated behaviour to obtain the recovered parameters. The mean and standard deviation of the non-decision time (NDT, rightmost plots) were sometimes confused (a larger mean substituted for a smaller variance), but the recovery of the behaviourally relevant process parameters (weight, urgency, and decision noise) was acceptable.”

RIC7

- Along the same lines as above, we are curious about the beta parameter that reflects the weight of evidence variability in confidence control (controlling over/under confidence). Are participants with better metacognitive sensitivity (Fig 1E) more calibrated with respect to this parameter???. It could be an extra way to validate the generative process of the model.

We thank the reviewers for this constructive suggestion. The beta parameter allows for under/over confidence in online confidence control, and so should contribute to metacognitive bias, rather than metacognitive sensitivity. That said, there are also other factors that can contribute to metacognitive bias in post-decision confidence ratings, so we hadn't thought to run this analysis (we assumed effects such as personality traits may make a stronger contribution as to how confident on average people report to be). Nonetheless, we performed a correlation between the beta parameter and confidence bias (measured as the median confidence across all trials) and found a marginally significant effect in the expected direction ($r = -0.48, p = 0.043$, left figure below). However, we only see this after removing two participants with rather extreme beta parameters (right figure below). Given that between-subject correlations can be dubious with relatively small sample sizes, and given our p-value is only marginally significant, we prefer to leave out this analysis but we would also consider moving this in the Supplement if the reviewer feels this is potentially useful.

RIC8

- The red line in Figure 1D is the average confidence in errors, whereas everything else is the median confidence. It should be labeled as the median confidence in errors.

We thank the reviewer for pointing this out, we now include a key on the Figure and have clarified the description in the legend P4L108:

“d) Median confidence in each condition, thin grey lines show individual participants, thick red line shows the average median confidence of incorrect trials, black line shows the average of all trials, shaded region shows 95% within-subjects confidence (barely thicker than black line).”

RIC9

- You perform t-tests between all conditions to show differences. We believe a proper post hoc test after ANOVA to correct for the number of tests conducted would be better.

We thank the reviewers for this suggestion, we now only highlight the cells of the post-hoc t-test table that would survive a Bonferroni correction ($p < 0.0014$, for 36 tests). We agree that for between-subjects or mixed designs,

Turkey's or Holm's methods are preferred. However, here we have a purely within-subjects design and so we do not lose too much power using post-hoc t-tests with a Bonferroni correction. We copy below Supplementary table S1 for demonstration.

	$p \rightarrow$ $t(18) \downarrow$	1.87E-03	3.47E-03	3.06E-05	7.80E-08	6.24E-10	2.67E-07	1.35E-07	2.63E-11
		3.555	0.783	0.024	8.24E-07	6.42E-06	4.02E-06	1.47E-06	1.43E-10
		3.292	0.280	0.006	4.44E-07	2.87E-06	2.57E-05	7.37E-07	2.19E-09
		5.286	2.424	3.080	1.39E-04	8.20E-05	7.50E-04	6.34E-06	9.43E-09
		8.027	6.892	7.182	4.645	0.829	0.313	0.612	8.07E-07
		10.662	5.964	6.321	4.868	0.218	0.325	0.925	4.94E-06
		7.423	6.171	5.360	3.940	-1.034	-1.008	0.107	4.85E-08
		7.756	6.625	6.944	5.969	0.515	0.095	1.683	2.43E-06
		12.677	11.569	9.930	9.123	6.901	6.080	8.266	6.398
									$\uparrow p$ $\leftarrow t(18)$

“Supplementary Table S1. Pairwise t-test statistics, effect of stimulus condition on discrimination performance (d'). Left of the diagonal provides the t-statistics, right provides p-values (two-tailed), with values less than the Bonferroni corrected significance level ($p < 0.0014$) highlighted in red.”

R1C10

- In the results of the tests, the F statistic is larger in confidence when the a priori effect seems smaller.

This is partly driven by confidence in the low-evidence condition, which largely and consistently differs from confidence in the other conditions. The other factor is that there is more between-subject variability in behaviour than was obvious from the figure showing the average proportion correct and response times with within-subject confidence intervals. We now include a subfigure showing individual participants proportion correct and response times as Figure 1e.

“Proportion correct by median response time in each condition (colours correspond to conditions in b) for individual participants.”

A side note, the two participants with the much shorter response times across conditions (dots to the left in this figure) are the participants with the extreme beta parameters we discuss in R1C7 above.

R1C11

- The way average confidence is reported seems odd; subjects cannot go back on their report.

We thank the reviewers for pointing out the lack of clarity with the respect to the confidence report. We now clarify in the Methods that participants could return to a lower confidence report in a second (or third if need be) revolution of the dot around the dial P13L414:

“The marker moved at 1 cycle per second, and always started at low confidence, with unlimited revolutions to get the desired confidence report.”

We also include a remark on this in the legend of Figure 1a P4L97:

“...it was possible to make more than one revolution to return to a lower confidence.”

R1C12

- In line 338 of the discussion, I think it's poorly worded, could it be? It should be something like "at the motor bound" instead of "that the motor bound"??

Thank you, we have corrected this typo as suggested (P11L323):

“...motor processes play an active role in decision formation^{27,28}, ultimately determining the choice at the motor bound...”

R1C13

- We didn't understand what you mean in the methods in the 'task' part with 'To prevent motion adaptation, every second trial the decision axis was vertically upwards and otherwise vertically downwards' (lines 421-422).

We are grateful to the reviewers for their detailed reading of the manuscript, this lack of clarity was also pointed out in R3C10. On each trial the mean direction of dot motion was close to vertical (vertically downward or upward), and participants were asked to discriminate if it was slightly more left or right of vertical. When the decision axis was vertically downward, the stimulus appeared like falling snow, and for vertically upwards, the stimulus was rotated 180 degrees. We found while piloting the experiment with only vertically downward (snow) stimuli, this induced a strong sensory adaptation to downward motion. The waterfall illusion is a fun demonstration of motion adaptation: . This adaptation would reduce sensitivity to motion and so interfere with performance in the task. We found interleaving upward and downward stimuli resolved this issue. To clarify the statement, while also being concise, we have added a reference to motion adaptation, and added some further description of the stimulus in parentheses P14L424:

“To prevent motion adaptation (Van Wezel & Britten, 2002), every second trial the decision axis was vertically upwards and otherwise vertically downwards (where stimuli relative to the vertically downward decision axis appeared similar to falling snow, while vertically upward stimuli were rotated 180 degrees).”

The addition of the table with the mean direction values in Figure 1b, with thanks to your suggestion R1C2 above, should also help clarify that the direction of motion was close to vertical.

R1C14

- In the EEG methods, you sometimes mention a frequency of 256 and in other lines, you mention 250 Hz. Overall,

the methods are too concise regarding EEG data preprocessing; the authors mention an inverted decoding model without much further specification.

We thank the reviewers for these points. The downsampling to 256 Hz was an error, now corrected to specify 250 Hz. In addition, we have reformulated the description of the decoding analysis to highlight the specification of inverted encoding model P19L573:

“The decoding analysis of the EEG used an inverted linear encoding model to estimate weights on sensor activity to best predict variables of interest. The inverted linear encoding model^{72,73}, assumes the EEG sensor signals (Y) encode a linear weighting (w) of the variable of interest (X), estimated by ordinary least-squares:

$$\hat{w} = (X^T X)^{-1} X^T Y \quad (11)$$

The variable of interest is then predicted by applying these weights to held out data (10-fold cross validation):

$$\hat{X} = (\hat{w}^T \hat{w})^{-1} \hat{w}^T Y \quad (12)$$

We examined how the model-predicted (primary and motor) evidence accumulation might be represented in the EEG sensor signals using this decoding analysis. To do this, we simulated evidence accumulation in each trial, using the double-integration framework and the parameters fit to each participant. For each trial, we took the median of the simulated evidence accumulation traces that predicted the observer’s behaviour (same choice made within 100 ms of their response time). Both the simulated primary and motor accumulators (for the chosen option) tended to increase over time and were strongly correlated over time (average Fisher transformed correlation $z = 1.02$). This correlation was broken by taking the derivative, where the slope of the primary accumulator in the lead up to the response is much flatter than that of the motor accumulator ($z = 0.23$). To match the sampling frequency of the evidence (120 Hz) with the sampling frequency of the EEG (250 Hz), the slopes of each EEG sensor were calculated over a 6-sample window (linear regression coefficient; ~ 42 Hz), while the slope of the simulated evidence was calculated over 3 samples (40 Hz). We found that applying a lowpass filter (8 Hz) on the EEG data prior to this stage improved the reliability of the decoding analysis (a comparison of filter choices is shown in **Supplementary Figure S4**). We then applied the above decoding analysis to estimate the weights (w) on EEG sensors (Y) that predict the simulated evidence (X) in the 400 ms window prior to the response (a comparison of windows of 300 and 500 ms is shown in **Supplementary Figure S4**). To maximally distinguish EEG signals corresponding to the primary and motor evidence, weights were estimated simultaneously (with an intercept term, X and w are of size $n \times 3$).

The precision of the decoding can then be assessed by taking the correlation between the EEG-predicted evidence and the model simulated evidence (applying a Fisher transform for normalisation, for the primary accumulator, mean $z = 0.024$, $t(18) = 5.90$, $p = 1.10 \times 10^{-5}$; for the motor accumulator, mean $z = 0.086$, $t(18) = 9.07$, $p = 2.47 \times 10^{-8}$). The precision computed separately at each sensor gives an indication of the topography of the contributions to the EEG representation.”

Reviewer #2 (Remarks to the Author):

This is a behavioral & EEG study which explores the role of confidence in online control of the decision process. The question is important because agents need to adaptively regulate the integration of evidence to perform optimally in dynamic environments. The model is quite elegant, especially the distinction between primary and motor accumulators which could resolve a number of debates and contradictory findings in the field.

However, I had some concerns about the model comparison and broader claims of superiority over previous models. I also found the neural evidence not very convincing, and at times felt like the paper would be better without it, even if this weakens the overall impact. But I am open to counter-arguments, and hope that the comments are found to be useful for improving the manuscript.

We thank the reviewer for their insightful and constructive comments, which have been incredibly helpful for improving the manuscript. To ensure we have fully addressed all comments, we have numbered each comment according to reviewer number and comment number, such that R2C1 corresponds to the first comment of this review. We provide a track-changed document with all changes commented with this code. In responding to reviewers, we quote the text, referring to the page and line number of the clean (not tracked-changed) revised manuscript, should the reviewer wish to find the changes in the clean manuscript also.

R2C1

Major comment (1):

It is reasonable to claim that the classic model is “unable to describe both how participants could be faster and more accurate with early high-quality evidence while also slowing down with early low-quality evidence.” But I don’t understand how it cannot even fit the red and lavender conditions, where evidence strength is stable. More generally, what constrains the classic model to such a tiny range of RT (0.93-0.99s) rather than spanning the full empirical range, which after all is bracketed by conditions that are not the critical conditions of the experiment?

I assume it’s because the evidence-varying conditions are interleaved with the stable ones, and perhaps the fit shown in 3A is quantitatively better than a hypothetical one which spanned the full RT range but missed the middle data points. (Or maybe there were there fewer trials of the stable conditions, so those would have less leverage on the fit?) But if so, this intuition is not spelled out, so it’s easy to be skeptical that alternative models were given a fair shake.

We thank the reviewer for highlighting this important point. The classic model can only adjust the response times by a) adjusting the bound (overall increasing or decreasing response times) or b) adjusting the noise parameter (affecting the spread of response times to, for example, increase the relative likelihood of later responses; but this also affects decision accuracy). The fit reflects the best the classic model can do with a single bound given the presented stimulus evidence: get the spread of accuracy as close as possible while approximately targeting the median response time across all conditions.

One constraint is, as the reviewer remarks, the fact that the conditions were intermixed, which means only one bound can be implemented. All conditions had the same number of trials (100 per participant), as outlined on P14L432. In supplementary Figure S2D we show how the bounds on a single accumulator would need to vary (by stimulus and over time) in order to capture behaviour (which is described in the main manuscript under the results, edited in response to R1C3, P7L193, copied below).

“**Supplementary Figure S2. Comparison of models and parameters....** D) Effective bounds on Primary accumulated evidence for one example participant, calculated by taking a running average primary evidence accumulated up to the time of decision commitment, separately for each condition. The figure suggests that if the participant were to implement behaviour suggested by the double-integration framework but with only a single-integration, bounds would need to not only differ across time, but with the evidence uncertainty.”

P7L193: “Describing behaviour in this dynamic sensory environment requires a model where decision processes can react to the unpredictable transitions in the evidence quality online (**Supplementary Figure S6**), which is provided by the double-integration model with online confidence control (**Figure 3b**; and would otherwise require effectively stimulus-time-dependent bounds on the primary evidence^{13,14}, **Supplementary Figure S2**).”

The other constraint, which was not so obvious in the main manuscript, was placed on the signal-to-noise ratio itself. Many evidence accumulation models estimate the ‘drift rate’ (signal-to-noise ratio of the evidence being accumulated) by fixing the noise standard deviation to some arbitrary value and fitting parameters to estimate the signal strength. Our stimulus gives us tight control over the external signal strength, such that we can measure the input frame by frame (variable v , in Figure 2a, described P5L127), which is disrupted by internal noise in the perceptual processing, with the standard deviation of the (gaussian distributed) noise fit as a free parameter in our model. This way we have 1 free parameter to describe the signal-to-noise ratio (drift rate) across all 9 conditions (actually, all 360 individual pre-generated stimuli) in the experiment. Whereas, if we were to fix the noise and estimate the signal, this would require at least 6 free parameters. All models in our analysis shared this constraint, so this approach is validated by the fact that double-integration model can describe behaviour under this constraint. The constraint is also justified by a separate GLM analysis (Supplementary Figure S1D) which shows the strong ability to predict behaviour based on the externally presented evidence independent of the computational model implementation.

It is possible that using 6 free parameters to describe the stimulus evidence (ignoring what was presented on the screen) could improve the apparent fit of the classic model, but this approach would suffer from a lack of parsimony (making the comparison with the more parsimonious double-integration model unfair). Rather, we attempted to weight the evidence in the single accumulator model in a similar way to how it is being weighted in the double-integration model (Supplementary Figure S6, copied below; except this does not come with noise filtering, which is not possible in the single-integration framework). Although this improved the fit, the fit was still worse than that of the double-integration model with confidence control. Moreover, the topography of the EEG associated with this weighted single-integration did not fit well with the existing literature, since it attributes decision processes almost entirely to EEG motor signals (see P18L529 in main manuscript, Supplementary Figure S6).

“Supplementary Figure S6. Examining possible bottom-up mechanisms. The double-integration model with confidence control (left panel of **A**) used an estimate of the variability of the decision evidence in computing confidence, which then affects the leak in the motor accumulator. We examined whether a weighting on the variability of the decision evidence could explain behaviour in a bottom-up manner, that is, affecting the primary evidence as it is accumulated in a single-integration framework. Note that the variability of the stimulus evidence is already taken account of in the decision evidence (Equations 1 – 3 in the manuscript), so this would amount to an over/under-weighting of the variability at the decision level. Using an additional free parameter, we fit the classic single-integration model with a weight on the decision evidence variability as in Equation 9 of the manuscript. This did improve the fit to behaviour (middle panel of **A**) relative to the other single-integration models, but BIC was larger ($\sum \text{BIC} = 83\,385$, see Supplementary Table S4) due to the extra parameter. Overall, the model overestimated performance while showing relatively improved estimates of response times. Allowing non-linear bounds in addition (right panel of **A**) also showed some improvement, but again the additional parameters did not make the fit parsimonious ($\sum \text{BIC} = 83\,403$). Neither model was found to be superior to the double-integration model with confidence control. Still, we questioned whether the latent model processes could possibly provide a better description of the EEG, given they rely on the single-integration process traditionally associated with central-parietal positivity. On the contrary, in order to fit behaviour, the classic model with the bottom-up weight estimated evidence accumulation more similar to the motor accumulator of the double-integration model, both numerically (average correlation in accumulator traces, $r = 0.9$) and in terms of the topography of the associated EEG signatures (**B**, left, vs **C**; though these signatures were less predictive of the latent processes, $z = 0.044$, compared to $z = 0.091$ for the motor accumulator in the double-integration model). The latent processes of the classic model with time-varying bounds and with bottom-up weight were localised to occipital regions (**D**), which is incongruous with the literature on neural signatures of evidence accumulation (and also not so well predicted, $z = 0.058$). One could also consider that even if the bottom-up models did produce good EEG predictions that made theoretical sense, it would be difficult to compete with the double-integration model in capturing EEG signatures across the full extent of the two accumulators (both central-parietal and motor signatures). Further, we showed in the manuscript that the double-integration model also provides a good prediction of post-decision confidence, whereas single-integration models require additional mechanisms to predict confidence (such as ongoing accumulation to a secondary bound). While bottom-up explanations may be appealing in terms of their mechanistic simplicity, the double-integration framework could be considered more comprehensive in terms of the additional behaviour and neural processes it encompasses.”

In summary:

The classic model fails because it has to use the same bound across all conditions, and because it is constrained by the stimulus evidence.

We show that the stimulus evidence is not an unfair constraint:

- a) The double-integration model deals well with this constraint
- b) A model-free GLM analysis shows behavioural responses can be predicted by the stimulus evidence

We do make several attempts to give the classic model a fair shake:

- a) By fitting time-varying bounds
- b) By using a similar weighting strategy as the double-integration framework, but prior to integration (though the temporal smoothing is not possible)

To try to make this clearer in the manuscript, we have added some of these details in the results, when first describing the fit of the classic model P7L187:

“The model was unable to describe both how participants could be faster and more accurate with early high-quality evidence while also slowing down with early low-quality evidence. This failure is in part due to the fact that the model has to implement a single decision bound across the intermixed stimulus conditions, and in part due to constraining all models to use the evidence from the presented stimuli (a constraint validated by a separate GLM analysis, **Supplementary Figure S1**). Similar model failures were also apparent for a variant of the single-integration model with time-varying bounds, and for a static LIT model (as implemented in previous work) where double-integration occurs with a smoothing parameter that is stable over time (see **Supplementary Figure S2**).”

R2C2

Major comment (2):

The key neural result is the correlation between corresponding traces in Fig 4A, bottom left, and Fig. 4B, left. But other than the lavender trace being above the others for the latter two-thirds of the graph, I don't really see a correlation here. And in the first third, the ordinal relationship is largely reversed across the two plots.

Perhaps there could be a better way to display this finding? One could try reducing the number of traces (e.g. collapsing to 4 conditions, as suggested above), or showing the pairwise time series data that make up the correlations, using a scatter plot. But as it stands I'm not sure the result is compelling enough to support a claim about neural mechanism, notwithstanding the interesting anatomical segregation of motor vs. central-parietal.

We entirely agree with the reviewer that the correlation in the accumulation traces across conditions is not so convincing. This is due to noise in the EEG data, as well as noise affecting behaviour (which means there is variability in the individual trial estimates of accumulation traces). The analysis aims to resolve the profile of the accumulation traces over time for individual trials rather than conditions per se. The variability across conditions is demonstrated by plotting the individual trial evidence accumulation traces, with the condition means, for two participants:

We were plotting the data by condition to be consistent with the convention of the other figures, but we would not expect a nice alignment between the EEG-prediction and the accumulation traces condition-to-condition because of the combined trial-wise variability (EEG-noise + behaviour noise) that makes the conditions less distinct.

We have implemented the suggestion of the reviewer (with thanks) to reduce the number of traces in the plots and offer a less noisy (and hopefully more convincing) picture of the relevance of the EEG signals in dissociating the theoretically derived response profile of the primary versus the secondary accumulator (i.e. linear vs quadratic, respectively). Specifically, we reduced to a single trace (collapsing across all conditions), which highlights how the overall rate of accumulation over time proposed by the model is well matched by the sensor activity. This also serves to reduce the space allocated to these figures, which hopefully helps redirect attention to the topography, which we think is most important. We have adapted the description in the main text in to highlight how the distinct shapes of the two accumulators is what should be appreciated P9L253:

“The overall shape of the EEG-predicted evidence accumulation profiles and those described by the model are well matched (**Figure 4B**), supporting the existence of these two distinct (linear vs quadratic) accumulation profiles in terms of neural correlates.”

“**Figure 4. Neural signatures of confidence control.** a) Accumulation profile (derivative of accumulated evidence) of the primary (left) and motor (right) accumulators from simulating the model fit to participants (black) and those predicted from the EEG inverted encoding model (red dashed line).”

More importantly, however, the key neural result we were interested in was actually the topography (anatomical segregation of motor vs central-parietal), which supports the neural implementation proposed at the theoretical level that the computational model attempts to describe. We have made this clearer in the results P9L257:

“The central purpose of this analysis was to assess the neurobiological validity of the proposed computational mechanisms, which can be appreciated from the topography of the sensor weights shown in Figure 4b: the motor accumulator most strongly relies on EEG signals over contralateral motor cortex; the primary accumulator is associated with topography comparable to a central parietal positivity signal, previously associated with evidence accumulation...”

R2C3

Major comment (3):

207-214: It’s impressive that the final state of the model-simulated online confidence predicted observers confidence reports, but I don’t think it’s fair to say that the current model offers a more natural explanation for confidence ratings compared to previous work. The current model gives an expression (Eq 8+9) for confidence but no plausible mechanism for reading it out, whereas previous studies offer a variety of mechanistic or process models (perhaps heuristic, yet often approximately optimal) for post-decision confidence.

Also, “since the bound on the primary accumulator implies the same quantity of evidence for all choices” seems incorrect to me. In a 2D accumulator, as in Kiani et al. 2014 and here, the state of the losing accumulator varies across trials, providing an explanation for variability in confidence (incorporating elapsed time) despite a fixed

value of accumulated evidence for the winner.

This is an important perspective which we are grateful to address. Framed this way, we agree with the reviewer that the term ‘natural’ is not so appropriate. We have removed it from the paragraph on P6L174 and also clarified that the mechanism by which the final confidence level is read out, remains open:

“Importantly, this framework also has the potential to ~~more naturally~~ account for post-decision confidence. Confidence is computed from the primary accumulator, which remains unbounded and is allowed to persist past the response. This ~~naturally~~ captures how confidence can incorporate additional evidence accumulated after decision commitment (without having to propose a double-bound, as in single-integration frameworks^{36,37}), although without specifying a read-out mechanism. Moreover, the smoothing in the motor accumulator suggests the decision is based on evidence affected by less noise than was present in the primary accumulator, explaining how confidence judgements reflect more intrinsic noise than decisions⁴¹ and accounting for the dissociation between confidence and decision sensitivity⁸.”

We have also changed the sentence that was referred to at 207-214 to P8L213 (also taking R3C2 into consideration):

“This validates the claim that post-decision confidence can be accounted for within the double-integration framework, and shows the latent model variable is related to participants’ explicitly reported confidence following their decisions.”

We were using the term ‘naturally’ to highlight three properties of post-decision confidence, which are commonly considered in the literature, that can be accommodated in the double-integration framework without additional parameters: 1) Confidence can vary over trials (similarly considering the state of the losing accumulator as in Kiani et al., 2014, or the ‘balance of evidence’ as termed in Vickers, 1979, or perhaps Audley, 1964); 2) Confidence can incorporate additional evidence not used for the perceptual decision (as the primary accumulator is unbounded; previously accounted for by specifying a secondary bound, Desender et al., 2022, or temporal cut-off, Pleskac & Busemeyer, 2010, for additional accumulation); and 3) Confidence is affected by additional noise, dissociable from perceptual noise (meaning variability in ‘M-ratio’, metacognitive sensitivity relative to perceptual sensitivity, which is accounted for here by the noise filtering in the motor accumulator).

But we agree with the reviewer that the model does not specify a read-out mechanism per se. We have added some discussion of this P12L344:

“While the experimental evidence presented here supports the central role of online confidence estimates in controlling behavioural efficiency, the evidence is inconclusive as to the mechanistic relationship between online confidence estimates and explicit metacognitive confidence reports (evaluated post-decision).”

And refer to the important work of Kiani and colleagues P12L357:

“Post-decision confidence has also been shown to incorporate additional information and biases, such as information accumulated after decision commitment^{8,36,37}, information about deliberation time (Kiani et al., 2014) and motor preparation⁵⁰, biases toward confirmatory evidence⁵⁴, biases based on stimulus visibility⁵⁵ and attentional allocation⁵⁶, and developing detailed computational models of how these factors influence metacognition represents a long-term goal for the field⁵⁷”

R2C4

Major comment (4):

300-317: This sounds like a cool idea but it comes a bit out of nowhere, and I failed to see how the analysis adds to the story about confidence control. If pupil dilation reflects commitment, faster dilation would be associated with

faster responses regardless of confidence control. The link with beta power is also not well developed, and empirically weak (pupil data are too variable to trust the bottom-right of 4F). Nothing wrong with exploratory analyses, but I'm not sure this one logically supports the claim being made, even in principle.

We entirely agree with this assessment that the analysis is exploratory, and the added value is weak. This echoes comment RIC5, and we have decided to move the analysis to the supplementary materials along with the other pupil-related analyses originally in the supplement (Supplementary Figure S5). We originally included this analysis as an attempt to offer an account of the theoretically-proposed “motor leak”, which we think might be dynamically moderated by LC activity. In other words, rather than an index of choice commitment, pupil dilation might instead index the extent to which confidence controls the dynamics of the motor accumulator.

“**Supplementary Figure S5. Decision preparation signals.** **A)** Effect of response speed (top row) and confidence (bottom row) on pupil size, pupil derivative, and motor beta power. In all cases a median split was used, within participants. Shaded error shows 95% within-subjects confidence (difference between split). Grey shaded regions show significant differences at the group level. Motor beta power was computed from the power spectrum across frequency tapers from 16 to 32 Hz with 25% spectral smoothing, resolved using wavelet convolution in FieldTrip. Channels were selected for each participant as the channel with the greatest (negative slope on beta power in the response epoch (C3 for 8 participants, and FC1, C1, and FC3 for 5, 4, and 3 participants respectively). **B)** Difference in the pupil derivative (velocity of change in pupil size; black) and beta power (red) using a median split of response times (fast – slow RTs). To make beta power better comparable with pupil size, the y-axis is reversed, such that decreased beta power corresponds to increased pupil size, the direction thought to indicate response preparation. Shaded regions show the 95% between-subjects confidence intervals. Thick lines at $y = 0$ mark regions of cluster-corrected significance for the difference in pupil velocity (black) and beta power (red) based on the median split of response times. Fast responses were associated with faster pupil

dilation following stimulus onset (cluster corrected significant window, 0.125 to 1.15 s; mean $t(18) = 3.28$, mean $p = 0.018$). The same median split of response times showed an appreciably similar effect on motor Beta power (red line; cluster corrected significant window 0.28 to 0.79 s; mean $t(18) = 2.75$, mean $p = 0.018$; the effects in the response-locked epoch also follow a similar shape, but the pupil derivative is too variable to reach significance). While preliminary in nature, this finding is in line with the hypothesis that motor leakage is implemented via subcortical inhibition, reflected in part in pupil dilation and resulting in decreased cortical motor Beta power with the release from inhibition.”

Minor comments:

R2C5

57-60: This part nicely situates the work relative to Zylberberg et al. and Levi et al., though I might suggest adding a nod to Drugowitsch, Moreno-Bote, and Pouget (NeurIPS 2014, Optimal decision-making with time-varying evidence reliability) if the authors agree it is relevant.

Thank you for pointing this out. In fact, we had intended to refer to this paper (which has served as inspiration for the current work) earlier at line 22 but instead entered a reference to Drugowitsch, J., ... & Pouget, A. (2019). Learning optimal decisions with confidence PNAS. We have fixed the earlier reference and added another nod at P3L56:

“In these scenarios, the observer would benefit from adapting decision processes online (Drugowitsch et al., 2014), to capitalise on early high-quality evidence, or down-weight early low-quality evidence in case better evidence follows.”

R2C6

102: Behavioural performance?

Good catch, actually we intended to write “Behaviour in a dynamic sensory environment”.

R2C7

140: At this point I was reminded of the distinction made by Pouget et al. (Nat Nsci 2016) between confidence and (un)certainty. The former is conditioned on a choice ($P(\text{correct} | \text{choice, evidence})$) whereas the latter is basically the inverse variance (or entropy) of the posterior over the stimulus dimension (or category). It sounds like the current study wants to claim there is a provisional degree of confidence calculated at every time point (Eq. 7), but then this is modulated by the variability of an additional noise term (Eq 8) incorporating a weighted estimate of the cumulative variability of the evidence — which sounds more like uncertainty, by the Pouget definition. (See also Li et al., Optimal policy for uncertainty estimation concurrent with decision making, Cell Reports 2023.) Of course the authors are welcome to call it whatever they want, but I think it matters whether the data actually support the idea that the brain continuously assigns a degree of confidence to a provisional choice at each moment, versus merely estimating the variability of the evidence (+ noise), as these two accounts may point to distinct neural mechanisms and make different predictions for subsequent work. Any clarity on this topic would I think be useful to add, perhaps in the Discussion.

We thank the reviewer for bringing up this intricate discussion. In line with Pouget et al., 2016 (and consistent with Kiani and Shadlen, 2009) our model proposes the primary accumulator encodes the log posterior distribution over choices (Eq 3), in this way corresponding to a certainty representation. Confidence in our model is conditional on a choice (Eq 7) which is in line with the definition of confidence as a particular form of certainty (as proposed in Pouget et al., 2016). Two aspects muddy the definition: First, as the reviewer points out, the observer hasn't yet committed to a choice, so this is more like ‘provisional confidence’; Second, the model uses confidence in each choice to modulate each (respective) motor accumulator, and in this case (since the choice is either $\mu_\theta < \theta_0$ or $\mu_\theta \geq \theta_0$) this covers the full posterior distribution of possible stimulus values and so could be considered

something closer to the definition of certainty. Eq 8 actually places our confidence model more in line with the definition of confidence from Pouget et al., 2016, accounting for a “suboptimal step in a downstream computation”, as in the second to last paragraph of Pouget et al., 2016.

Specifically, Eq 8 allows the model to suggest the participant does not have perfect knowledge of their sensory certainty, a miscalibration in estimating their own noise. To be clear, both certainty and confidence are modulated by the variance of the distribution according to Pouget et al., 2016, since confidence relies on certainty conditioned on a choice.

To address this comment in the manuscript we have:

1. Adopted the term ‘provisional’ from the reviewer (with thanks) to better link the model description in the results to the definitions of confidence and certainty from Pouget et al., 2016, P6L149:

“With confidence modulating the leakage in the motor accumulator, the amount of smoothing now becomes a dynamic variable and corresponds to current certainty conditioned on the provisional choices.”

2. Added explicit discussion referencing the work of Pouget et al 2016 to the discussion, and speculating about possible mechanistic connections P12L344:

“While the experimental evidence presented here supports the central role of online confidence estimates in controlling behavioural efficiency, the evidence is inconclusive as to the mechanistic relationship between online confidence estimates and explicit metacognitive confidence reports (evaluated post-decision). Online confidence estimates used during deliberation could rely on representations of certainty from low-level (e.g. sensory) probabilistic neural population codes⁵¹. Post-decision metacognitive confidence reports have been associated with a read-out of this low-level distributed coding in high-level brain regions⁵² (frontal cortex). While our model formalisation of online confidence is consistent with this high-level read-out (conditioning certainty on the provisional choice), and we find the model online estimates predict post-decision confidence reports, the read-out for online confidence could rely on a distinct mechanism to that of explicit post-decisional confidence reports.”

R2C8

Fig 3: Having left Fig 1 behind I found it difficult to remember which color corresponds to which condition. It might help to reinsert Fig 1B here (and/or in Fig 4) as an inset. But a more radical idea is to regroup the conditions as, say, ‘stable-high, stable-low, early-high and early-low,’ since no key point is being made about changing the mean vs. variance. This would make all the graphs less spaghetti-like, including Fig. 4A+B and Supp Fig S2 which are very hard to parse. Just a suggestion.

We are very grateful to the reviewer for this suggestion, which is also helpful for addressing R1C2. We have chosen to reinsert Fig 1B as an inset for the following figures in the manuscript: Figure 2c; Figure 3a; Figure 4d. And the supplement: Figure S2A; Figure S6A.

R2C9

Fig. 3E: Why not show the unity slope lines, as in D?

We thank the reviewer for this suggestion, we have corrected the absence of unity lines in Figure 3e:

R2C10

241: What is the interpretation of the derivative of an accumulator? Is this just a way to maximize the differentiation between primary and motor, or is it because of the need to compare with the derivatives of EEG signals?

It is a just a way to maximise the differentiation between the primary and motor accumulators. The EEG decoding analysis is similar to a multiple linear regression, where we decode the two accumulators within the same analysis to maximally differentiate the neural signatures of these accumulators. A correlation in the variables of interest would disrupt this similar to multiple regression approaches. We outline this more thoroughly on P9L240:

“Due to the overall ramp profile in both accumulators, the raw primary and motor accumulators were strongly correlated (average Fisher transformed correlation $z = 1.02$) which would disrupt our ability to isolate separable EEG components simultaneously associated with these two accumulators. However, the derivatives (slopes, **Figure 4a**, bottom) show a weaker correlation ($z = 0.23$), as they distinguish the linear vs exponential profiles of the primary vs motor accumulators (respectively).”

R2C11

248: Speaking of which, what aspect of the signal is being extracted? Just the raw signal, or power in some frequency band? This relates to the above point that we are not given intuition for why the derivative is used.

We thank the reviewer for pointing out this lack of detail. We have now specified that this is the amplitude (“raw”, but after pre-processing and smoothing (with an 8 Hz lowpass filter) P9L249:

“...from the derivative of the EEG signal amplitude in the 400 ms leading to the response (see **Methods**).”

We do prefer to leave the majority of the details about the analysis to the methods, but ensure to point the reader there explicitly now.

R2C12

249: Why is this called precision?

We adopted the terminology of Salvador et al., 2022, who use this same metric. The inverted encoding analysis includes a bias/intercept term that aligns the mean of the prediction to the mean of the data (meaning the prediction couldn't be precise without being accurate). The metric itself is the Fisher transformed correlation between the predicted and actual variable.

R2C13

Fig 4C,D: Not everyone is familiar with these EEG array plots; could you add a contour/boundary to illustrate the

location of motor and parietal cortices (determined independently from the activity heat map)? Also, there's no legend for what red vs blue means.

We thank the reviewer for these helpful suggestions, which we have implemented in the revised Figure 4b and c:

“**b)** Topography of the EEG encoding of primary (left) and motor (right) evidence accumulation (Fisher transformed precision increases in orange, white marks the half-maximum). The green marker shows sensor C3 (commonly associated with right-hand motor responses) and the purple marker shows sensor CPz (over which the central parietal positivity signal is centred). **c)** Topography of the EEG representation of post-decision confidence (same format as **b)**.”

R2C14

293: Only the last 200 ms are used, understandably, but confidence control (in this task) is required from trial onset. Is the assumption that the strength of confidence control fluctuates across but not within trials, such that estimating it using the final 200 ms is sufficient to apply that to the whole trial?

We use the final 200 ms because this is where the ramp in the motor accumulator is most pronounced. Prior to that the motor accumulator derivative is smaller and so makes the analysis more vulnerable to noise. This is the primary reason we note on P10L293:

“We measured the endogenous variability of the strength of confidence control using the within-trial correlation between the EEG-predicted online confidence and the ramp-up of the EEG-predicted motor accumulator, in the 200 ms prior to the response (where the ramp is most pronounced, and prior to which the smaller ramp makes the analysis more vulnerable to noise).”

In addition, we cannot perform the analysis using data locked to stimulus onset, because the ramping of the motor accumulator is associated with response time itself, and so the results would be confounded by response time.

R2C15

310, 314: Oddly coincidental that both of these p values are exactly 0.018. And is it valid to make a conclusion of significance based on the mean of multiple p values?

We too noted this coincidence when first making the summary (and double checked). Of note, the conclusions are not based on the mean of multiple p-values, this is just one way to summarise the statistics of the cluster. The conclusions were based on the cluster-correction approach of Maris and Oostenveld, 2007. In line with the suggestions R1C5 and R2C4, we have moved this analysis to the supplementary materials.

R2C16

392: direct -> direction

Yes, thank you.

R2C17

426: Were Ss informed about the dynamics of stimulus strength? Or if you asked them afterwards, did they report that they detected that something was changing in the first second? I think it's worth noting/speculating whether the predictable timing of the transition at $t=1s$ had anything to do with the clustering of RTs around that time (most between 0.85-1.15s, doesn't seem like an accident).

Participants were informed that the stimulus could become harder or easier during stimulus presentation, but weren't told about the timing of these changes. The ability to estimate a duration of 1s is moderately biased (average is ~960 ms) and somewhat variable (coefficient of variation ~0.17; from Coull et al., 2013, Timing & Time Perception), but could also be affected by the presentation of the motion stimulus itself (e.g. Yamamoto & Miura, Vision Research, 2016) so it is possible participants could have considered a form of cut-off for deliberation according to some internal clock, if they became aware that the larger dynamics finished around 1s (there are still some dynamics after 1s as the dot directions continue to be re-sampled, though the generating distribution parameters remain the same). Participants were encouraged to try to respond faster if their median response time (across all conditions) was greater than 1.5s over a block of 90 trials, which could also have also helped to inform a deliberation cut-off. Participants were also told that if they saw the stimulus disappear (which occurred after 2s of stimulus presentation) they had taken too long to respond. This kind of deliberation cut-off would be captured in our model by the t_d parameter which determines the timing of a confidence independent increase in the motor leakage, simulating an urgency to commit to the decision (Eq 10, and P18L517).

We now present in Figure 1e the proportion correct and median response times of individual participants, to demonstrate the variability across individuals (not so evident from the within-subject confidence intervals in Figure 1c), which we hope reassures the reviewer that this apparent clustering of RTs around 1s is not systematic across individuals. We also present the full distribution of response times in **Figure 3c**, where the reviewer can appreciate there are a substantial proportion of response times greater than 1s, with no evidence for a strict cut-off which would imply a strong influence of the kind of heuristic cut-off described above. As such, we prefer not to add speculation about the possible additional influences on response times in the main manuscript.

“Proportion correct by median response time in each condition (colours correspond to conditions in b) for individual participants.”

“c) Probability density of responses by response time. Correct responses are shown upward of the 0-point, and incorrect, downward. Markers show participants, red solid line shows the prediction of the classic model, and black dashed line shows the prediction of the confidence control model.”

R2C18

There are two Supp Fig S5's (the 'bottom-up' one should be S6 I guess)

Yes, thank you for spotting this. We have amended the text accordingly.

Reviewer #3 (Remarks to the Author):

In this paper, Balsdon and Philiastides test the performance of a two-accumulator model of perceptual evidence in its ability to explain participants' decisions to respond, or to wait and accumulate further evidence before committing to a decision. They also seek neural correlates of these two proposed accumulators (finding that they are plausibly separate, in neural terms, and that the second may be associated with motor signals).

This is a fantastic study: interesting, timely, and creative in its experimental design. The analyses are sound, carefully conducted, and clearly explained. I congratulate the authors for their good work, that I thoroughly enjoyed reading and thinking about. This study is computationally very close to the own authors previous work, so I think some further clarification would help explain the conceptual novelty of their findings. I have, additionally, a few comments or questions.

We thank the reviewer for these encouraging remarks, and the constructive and helpful comments below. To ensure we have fully addressed all comments, we have numbered each comment according to reviewer number and comment number, such that R3C2 corresponds to the second comment of this review. We provide a track-changed document with all changes commented with this code. In responding to reviewers, we quote the text, referring to the page and line number of the clean (not tracked-changed) revised manuscript, should the reviewer wish to find the changes in the clean manuscript also.

R3C1

Major comments:

- Interpretation of Figure 1E: The authors hypothesised that participants with a tighter relationship between confidence and decision accuracy would also show a stronger modulation of the experimental manipulations. This is fine, but I can think of a few possible confounds for why this effect might emerge.
 - i. First, if I understood it correctly, the results of the GLM they used to measure the association between confidence and performance depends on the within-subject range of performances. I.e, a participant with a narrower performance range will have a less clear association with confidence. So it's not a given that this measure can be used to compare different participants.
 - ii. Also, (and I hate to be 'that guy', but it should be said) simple and general motivational or attentional effects could explain this relationship. A participant that is more attentive throughout the task will likely show a stronger effect of the manipulation and also have a better association with confidence. Attentional effects are often a trivial (and admittedly boring) suggestion as alternative interpretations, but the authors could perhaps make it clear that the effect they see is compatible with their hypothesis, but does not provide direct evidence for it, unless they can show in a control analysis that this effect is not general.

We are grateful to the reviewer for these alternative interpretations. We entirely agree (except for the part about attention being boring), this analysis was perhaps confirmatory from the outset. We note that these possible confounds should be considered within the mechanistic framework. Under a single-integration framework, possible confound i) would be consistent with a reduction in noise, leading to increased accuracy and increased confidence sensitivity, but reduced variability in response times (inconsistent with our findings); possible confound ii) would be consistent with more motivated/attentive participants increasing their decision bound (perhaps in addition to reduced noise), leading to increased accuracy and increased confidence sensitivity, but overall slower response times (inconsistent with our findings). Our findings show participants with better confidence sensitivity are able to increase their speed and accuracy for early high-quality evidence while slowing down for early low-quality evidence, consistent with better confidence control (but not consistent with changes in noise or bound placement as could be implemented in single-integration frameworks). However, we agree this does not count as direct evidence for online confidence control, so we have removed this analysis. The model comparison, supported by the prediction of post-decision confidence, and the EEG evidence supporting the neural implementation, provide sufficient evidence.

R3C2

- In line with my previous comment, I am not sure that the authors' interpretation of the signal c as 'confidence' is

granted, and it is definitely not shown. If I missed something, I apologise. The model poses that a readout from the first accumulator c is used to modulate the second accumulator. But the authors never test, for example, that the modelled final value of c corresponds to participants' reports of confidence. Indeed, I am a little bit surprised that the paper is so centred on the role of online confidence on decisions (even having it on the title) but there is so little emphasis on the analysis of participants' actual confidence reports. I couldn't access the preregistration (it's embargoed), but it would be good to know if the authors had planned any more analyses of participants' explicit confidence reports.

We thank the reviewer for this important comment. We do show the correspondence between the model confidence and the post-decision confidence ratings, this is what is plotted in Figure 3e. To make this more obvious, we have separated out the text that describes this analysis in its own paragraph, and make an explicit statement about validating the claim that the model latent variable estimating online confidence is related to post-decision confidence P7L204:

“As additional evidence of the superiority of the double-integration framework, we were also able to predict post-decision confidence ratings from the model fit only to choices and response times, without additional parameters. For each trial, we simulated 1000 noisy instances of the double-integration process, using the parameters fit to each participant, and took the median of the instances consistent with the participant’s choice and response time on that trial. This provided a trial-wise estimate of the dynamic primary and motor accumulated evidence, as well as the online confidence computed from the primary accumulator. The final state of the model-simulated online confidence for the chosen response predicted observers’ post-decision confidence ratings with $z = 0.41$ (average Fisher transformed correlation; range, [0.0604, 0.5634]; t-test against 0, $t(18) = 12.425$, $p = 1.45 \times 10^{-10}$) and captured the pattern of confidence across decision accuracy and response time (**Figure 3e**). This validates the claim that post-decision confidence can be accounted for within the double-integration framework, and shows the latent model variable is related to participants’ explicitly reported confidence following their decisions.”

“e) Participants’ confidence (z-scored) by confidence predicted from the model simulated confidence (using parameters fit to choices and response times but not confidence), across trials split by median response time (fast RT vs slow RT, left) and correct/incorrect discrimination decisions (right).”

We also changed the axis label to read ‘Model simulated z(confidence) instead of simply ‘Simulated z(confidence)’ to make this more explicit.

This analysis was pre-registered as an exploratory analysis (second-to-last section of pre-registration; apologies, we have now released the pre-registration from embargo):

“We expect some exploratory analysis will be required to examine the relationship between confidence and evidence accumulation across conditions. Initially we plan to do this by comparing model predictions of confidence from the simulated primary and motor accumulators”

In the manuscript we only present the model predicted confidence from the simulated primary accumulator. We thought it may be possible that post-decision confidence also incorporates some evidence from the motor accumulator (as suggested in Gajdos et al., 2019, *Neurosci. Conscious.*) but in our data the primary accumulator alone was sufficient. We still refer to this possibility in the discussion P12L341:

“While the framework allows for ongoing primary evidence accumulation after decision-commitment to contribute to post-decision confidence, further investigation is required to understand how other factors (for example, motor processes⁵²) might additionally be incorporated into post-decision confidence.”

We also have the EEG results, where we use the decoder predicting post-decision confidence to estimate online confidence and show that these online EEG confidence signals drive faster more accurate decisions as predicted by the model.

We did try to emphasise this part of the analysis in the discussion too, P11L333:

“We show that the EEG signatures of post-decision confidence can be used to estimate online confidence in a way that is meaningful for behaviour. Moreover, the final estimate of online confidence from the model fit to choices and response times provided a good prediction of post-decision confidence responses.”

We would be happy to receive any additional feedback about how to clarify and highlight this aspect of the manuscript further.

R3C3

- I am not an expert in evidence accumulation models, so I cannot judge how novel this model is. However, the authors already presented and tested this double accumulator model (e.g. ref 16). Here, the argument is that this model not only better predicts decisions in different experimentally-controlled conditions of speed-accuracy tradeoffs, but also explains participants' own cognitive control. That is fine, but is by no means a novel argument, as previous studies have shown that confidence drives decisions to seek further information (e.g. Desender *Psych Sci* 2018). In fact, these other studies have shown it less ambiguously, as it was confidence reports themselves, and not a latent variable that is assumed to correspond to confidence. Perhaps the authors would argue that the novel aspect is the volatility of the environment, and how much better the two-accumulator model fares as compared to a single-accumulator model. Either way, given how much (excellent) prior work the authors have done on this topic, it would be good, I think, to better explain in the discussion what new piece of the puzzle they have found in this specific study.

We are grateful for these suggestions from the reviewer which have helped us clarify the novelty of the current model. This model is not the same as we used in previous work. In previous work, the motor leakage parameter, which we found controlled the speed-accuracy trade-off, was static over time (a single value per experimentally controlled speed-accuracy trade-off condition). Here we allow the leakage parameter to be controlled by online confidence (and so it varies over time). This adjustment may seem like a technical detail, but it has substantial consequences at the theoretical and practical levels. At the theoretical level, it transforms the model process into an approximate Kalman filter, an optimal approach for estimations based on measures with temporally independent noise; we show in Figure 2c how this implementation can improve both the speed and accuracy of decisions (i.e. decision efficiency). At the practical level, it is this use of online confidence to control the motor leakage that serves to explain behaviour; as stated P7L190 and extended in Supplementary Figure S2, the static LIT model cannot explain behaviour in this dynamic environment. We now make it more explicit that by ‘static LIT model’ we are referring to the previous framework P7L190:

“Similar model failures were also apparent for a variant of the single-integration model with time-varying bounds, and for a static LIT model (as implemented in previous work) where double-integration occurs with a smoothing parameter that is stable over time (see Supplementary Figure S2).”

We also make the changes from the previous model more apparent when we first introduce the model P5L113:

“We formulated an extension of the Leaky Integrating Threshold model (LIT¹⁸) to account for online modulation of decision processes via metacognitive confidence control. The LIT model employs a double-integration framework which was initially motivated by neurobiological plausibility: the primary accumulated evidence is re-integrated by a secondary, motor accumulator, which triggers the behavioural response at its own bound. This accounts for the build-up of motor activity prior to decision commitment and formalises the more active role of motor processes in decision-making^{28,29}. This framework also allows for flexible online modulation of the evidence accumulation rate, via leakage in the motor accumulator. In previous work we showed how the model can capture changes in the speed-accuracy trade-off via changes in a static leakage parameter (henceforth referred to as the static LIT model). Here we formulate an extension by proposing that motor leakage is controlled by confidence, which is computed from the (unbounded) primary accumulator. Critically, this means online confidence control is implemented without circularity, since the evidence for confidence is decoupled from the final evidence for the decision.”

We also try to emphasise this change from the previous model more clearly in the introduction P2L37:

“This secondary accumulator has a “leaky memory” and uses this property to consider past evidence in the accumulated signal, rather than just its current state. Importantly, this property serves to modulate both the quality and quantity of decision evidence, which here we implement as being controlled by online confidence computed from the primary accumulator.”

We agree with the reviewer that the experiments presented in Desender et al., 2018 are very relevant, we have added this reference to the introduction P2L19 (we apologise for omitting this reference in the previous manuscript):

“Indeed, there is growing endorsement for a role of confidence in adjudicating how much evidence is required to commit to decisions (Desender et al., 2018; and refs 12-14), consistent with earlier proposals implicating a belief about obtaining a reward (cost-benefit trade-off^{15,16}). Here we test a framework in which confidence does not only control the quantity of decision evidence, but is used to actively improve evidence quality.”

As explained in the quoted sentence, our novel proposal is that confidence is not only used to control the quantity of evidence (as implied by decisions to seek more information), but also improve evidence quality (via the temporal smoothing of the motor accumulated evidence). The current manuscript therefore suggests a far more central role for online confidence than simply information seeking, and further, we provide a mechanistic description of this role with a neurobiologically plausible model. We noted this also in the discussion P11L319:

“Second, confidence control not only moderates the quantity of evidence accumulated (as provided by decision boundary shifts), but also the evidence quality (via the smoothing induced by motor leakage)”

While the experiments of Descender et al., 2018 and the one presented here show an effect of stimulus manipulations on reported confidence, neither ask for reports of confidence ‘online’ prior to further information seeking. Desender et al., 2018 show that stimulus conditions for which participants were more likely to seek further information were also associated with lower confidence on a separate set of trials where participants were not offered the opportunity to seek more information.

In addition to showing an effect of stimulus condition on reported confidence (P4L91), we do also show that the model latent variable assumed to correspond to online confidence is related to explicitly reported post-decision confidence on those same trials, as we highlight in response to R3C2 above.

R3C4

- Could the authors explain why the confidence was reported in the way it was? Could this have led to any kind of confound in that more motor preparation for a longer response correlated with higher confidence, hence appearing as a motor topography of the second accumulator? (I understand that this is 400 ms prior to the discrimination response, but also confidence is argued to be computed online, so perhaps the two are not separable).

We thank the reviewer for this comment. The lack of clarity regarding the choice of confidence report method was also highlighted in R1C1, and we have added further justification of this choice in the methods P14L416:

“This confidence dial design limited motor and eye-movement artefacts that could disrupt the EEG measures.”

We have also clarified that more than one revolution was possible, so participants could return to a lower confidence rating if they missed it on the first revolution. This means that the relationship between confidence and confidence response duration is not strictly determined. There is also considerable variability in the time participants initiate the dial response (relative to dial onset; see figure below), meaning that the timing of motor preparation for the confidence response is not aligned with the timing of the perceptual decision response. If there were EEG signals associated with the duration of a button press (though some evidence suggests this is not the case; McCone, Butler, & O’Connell, *BioRxiv*, 2023) these would not be systematically associated with the timing of the perceptual decision response, and so not disrupt our analysis locked to the timing of the perceptual decision response. We note this variable confidence response execution time now in the legend of Figure 1 P4L96:

“After 200 ms observers were cued to make a confidence rating, entered after unlimited time by holding a response key to move the circular marker around the annulus until it reached their desired confidence”

In addition, our decoding analysis revealed neural signatures of post-decision confidence with a central-parietal topography, consistent with previous studies. While there is some overlap with the motor accumulator topography, the confidence topography is much less lateralised, inconsistent with being driven by motor planning.

Here, confidence RT is the time from the confidence dial onset to the time the participant entered their confidence response (button release). The lower bound is determined by the duration of the dot revolution around the dial (1s per revolution), the additional variability is due to the time prior to commencing the dot revolution (pressing down on the button) as well as performing multiple revolutions to return to lower confidence. Each marker in the figure is one trial from one participant; all trials with confidence RT less than 2 s are shown.

R3C5

- Related to the comment above, a more general/conceptual question: Do the authors speculate that this second

(motor) accumulator would be implemented differently in neural terms, if the confidence response were provided differently? My reading of the paper is that, because confidence is used online for decisions to commit to a decision or wait, it would exist even if participants hadn't been asked to rate confidence at all. Is this what the authors argue? I think that if the authors really want to make such a strong claim about confidence, this might be a possible control experiment that could help their argument.

We thank the reviewer for this interesting question. There are many ways confidence can be reported. In ecological contexts we are more likely to use verbal descriptions to communicate to others how much they should trust our decisions. We think online confidence should be used to improve decision efficiency irrespective of whether the observer plans to report their confidence, and irrespective of their method of doing so. We now state this explicitly in the discussion, and highlight that additional experiments could test for this P12L353:

“We also propose that online confidence is used in decision-making irrespective of whether an observer expects to give an explicit report of post-decision confidence, and independently of how post-decision confidence is reported. These reporting factors could influence the read-out mechanism for post-decision confidence, while online confidence should follow the ideal computation for the decision at hand (though this remains to be tested).”

The reason for this (and this relates also to comment R2C7) is it remains entirely possible that different mechanisms are involved in online confidence and post-decision confidence reports, while they both rely on the same ‘certainty’ information. We do show that EEG signatures of post decision confidence can be used to obtain estimates of online confidence that are behaviourally relevant P10L292. But indeed, this overlap in EEG signals could be explained by two mechanisms using the same certainty information. The focus of the current work is on the use of online confidence control, we leave the relation between online confidence and post-decision confidence report (and indeed their mechanistic implementation) to future studies.

To clarify, the model does not formalise the role of online confidence as deciding when to commit to a decision. Confidence is used to apply the appropriate temporal filtering to measurements with temporally independent noise (control the leakage when re-integrating the primary accumulated evidence). Incidentally, this also appropriately speeds up or slows down decision commitment.

R3C6

- The two-accumulator model is very different from recent accounts (admittedly in detection contexts) suggesting that a single accumulator model is enough to explain confidence as the difference between maximal evidence and the distance-to-bound. (Pereira et al, Nat Comms 2021). Perhaps the changing signal-to-noise ratio in the task shown here is enough to explain the difference, but I was surprised to see no reference at all to this work.

We thank the reviewer for raising this point. The purpose of the current work is not to propose a competing explanation of post-decision confidence. The focus of the current work is in describing a neurobiologically plausible role for online confidence in decision-making. Our interest in post-decision confidence here is more to validate that the model estimates of online confidence are at least somewhat related to explicitly reported confidence. It just turned out that the final state of the model estimated online confidence happened to be quite a good predictor of post-decision confidence (without the need for additional parameters or modifications).

We believe many of the developments in recent work on post-decision confidence can be implemented within the double-integration framework. Indeed, there are many single-accumulator models that can sufficiently account for post-decision confidence in various ways (there are also very good non-process models; 14 models are compared in this paper: Shekhar, M., & Rahnev, D. (2023). How do humans give confidence? A comprehensive comparison of process models of perceptual metacognition. *Journal of Experimental Psychology: General*).

The formalisation of online confidence in the current model is not in direct conflict with this previous work on post-decision confidence: confidence is still based on the primary accumulator (which corresponds to the single-accumulator). How confidence should use the evidence from the primary/single accumulator could be task dependent. Many models require additional evidence accumulation, surpassing the bound for the first-order

decision, to explain post-decision confidence (this is certainly the case for the model of Pereira and colleagues, 2021, as the bound forms part of the confidence equation, which makes some sense for detection tasks).

There are also many aspects of post-decision confidence that are not (yet) built-in to the current double-integration formalisation of post-decision confidence, we now highlight some examples of this in the discussion P12L357:

“Post-decision confidence has also been shown to incorporate additional information and biases, such as information accumulated after decision commitment, information about response time and motor preparation, biases toward confirmatory evidence, biases based on stimulus visibility and attentional allocation, and developing detailed computational models of how these factors influence metacognition represents a long-term goal for the field.”

We mention, in particular, the long-term goal, because no current framework is built to deal with all these issues. It's possible the double-integration framework could help unify some existing approaches, or perhaps it is the very formalisation of the computation for confidence that needs to be clarified (for example, whether a more heuristic computation is more appropriate).

Minor comments:

R3C7

- Line 19: I would suggest to remove the motivation to study this based on the (perhaps simplistic) assumption that of what would and not be inefficient for the brain to do.

We thank the reviewer for this suggestion, we have changed the sentence to P2L18:

“However, there is increasing experimental support for signatures of confidence emerging online, during decision-making⁸⁻¹¹, suggesting confidence could serve an active role prior to decision commitment.”

R3C8

- Figure 4.A: are the colours mapped wrong? Would one not expect the purple line to accumulate evidence slowest?

This was due to the accumulation traces being aligned to the time of the response: By the time of the response the observer has (on average) already accumulated more evidence in the low evidence strength condition (accumulated evidence over a longer duration), and it is also more likely that these responses were driven by noise (which on average looks like an increase in accumulation speed) as evident from the lower proportion correct. Thanks to the suggestion of R2C2, we have changed this figure to show just the overall evidence accumulation profiles (not separated by condition) so this should no longer present confusion.

R3C9

- Line 417: minor typo? "More to the leftward or rightward of vertical" seems grammatically wrong.

Yes, thank you, we had changed this to: "... moving more leftward or rightward of vertical..."

R3C10

- Line 421: I did not understand the difference between vertically upwards or downwards. Was the decision not based on a full vertical midline of the stimulus?

We thank the reviewer for this comment, which was also brought up in R1C13. We hope to have clarified by adding a more vivid description of the stimulus P14L424:

“To prevent motion adaptation⁶³, every second trial the decision axis was vertically upwards and otherwise vertically downwards (where stimuli relative to the vertically downward decision axis appeared similar to falling snow, while vertically upward stimuli were rotated 180 degrees).”

The suggestion of RIC2 to add a table describing the conditions perhaps also helps in establishing early on that the mean dot direction was always very close to vertical (not horizontal), which we describe in more detail in the legend P4L98:

“The quality of the sensory evidence was manipulated according to nine conditions created by adjusting the parameters of the circular Gaussian (von Mises) distribution from which the dot directions were sampled. The table to the left shows the parameter values in each condition: the mean difference from vertical (degrees) and the Kappa parameter which controls the inverse spread of the distribution (arrows highlight the change in terms of evidence strength, which is also depicted to the right). In four conditions the evidence was systematically increased or decreased over the first second of stimulus presentation (by increasing/decreasing the mean or the variance). In three conditions the evidence was completely stable. Two additional conditions of moderate evidence were created by changing the mean and variance in opposite directions.”

Reviewer #4 (Remarks to the Author):

We are very grateful for the time and effort put into this review, as well as for the highly constructive and helpful comments. We hope this training experience has also been positive for the reviewer.

REVIEWER COMMENTS

Reviewer #1 (Remarks to the Author):

All our concerns have been addressed, we congratulate the authors for this work

Reviewer #2 (Remarks to the Author):

I thank the reviewers for so thoroughly and thoughtfully addressing every single comment. It's really impressive and adds to my high regard for the quality of the work. The paper also reads better.

Still, I feel the need to push back just a bit regarding R2C1:

I understand the constraint of interleaved conditions, and of using the actual stimulus evidence rather than fitting arbitrary drift rates. But I still think the classic model is not fairly represented by the square data points in Fig. 3a (RT range 0.93-0.99). Using a simple 1D DDM simulation, within just a few minutes I was able to hand-pick parameters that roughly matched the accuracy range *and* RT range of your data (range see attachment). This simulation has three free parameters: a sensitivity term (k , multiplied by coherence to give drift rate), bound height, and non-decision time. Momentary evidence is normally distributed with mean = $k \cdot \text{coh}$ and s.d. held constant. Contrary to the rebuttal comment, this approach would not “require at least 6 free parameters”; one simply assumes that drift rate is proportional to evidence strength and you fit the k . [fwiw I used $\sigma = 0.8$, bound = 30, $t_{nd} = 100$ ms, and $k \cdot \text{coh}$ ranged from 0.003 to 0.023]

Again it's not the intermediate (dynamic) conditions I'm worried about. I'd bet 100 bucks that those will still strongly support the double integration model, especially given the 'concavity' of the overall pattern in Fig. 3a compared to convexity of my simulated data. Converting the dynamic evidence profiles to a presumed (latent) evidence strength that is fixed within a trial is a little tricky, since a simple average would have to specify up to what time. But assigning some stationary representation of momentary evidence (despite what's on the screen) is exactly what you'd want for a model that is explicitly designed as an alternative that does not care about dynamic evidence.

Anyway the details of the simulation shouldn't matter much — my only point is to demonstrate that a single accumulator can at least approximate the edge conditions, even with interleaving, and

constrained to use the presented stimulus evidence (because that does not vary for the edge conditions!), without compromising parsimony.

What am I missing?

Reviewer #3 (Remarks to the Author):

The authors have clarified all of the issues I raised, and modified the manuscript accordingly, when necessary.

I have no further comments.

Reviewer #4 (Remarks to the Author):

Reviewer #2 (Remarks to the Author):

I thank the reviewers for so thoroughly and thoughtfully addressing every single comment. It's really impressive and adds to my high regard for the quality of the work. The paper also reads better.

Still, I feel the need to push back just a bit regarding R2C1:

I understand the constraint of interleaved conditions, and of using the actual stimulus evidence rather than fitting arbitrary drift rates. But I still think the classic model is not fairly represented by the square data points in Fig. 3a (RT range 0.93-0.99). Using a simple 1D DDM simulation, within just a few minutes I was able to hand-pick parameters that roughly matched the accuracy range *and* RT range of your data (range see attachment). This simulation has three free parameters: a sensitivity term (k , multiplied by coherence to give drift rate), bound height, and non-decision time. Momentary evidence is normally distributed with mean = $k \cdot \text{coh}$ and s.d. held constant. Contrary to the rebuttal comment, this approach would not “require at least 6 free parameters”; one simply assumes that drift rate is proportional to evidence strength and you fit the k . [fwiw I used $\sigma = 0.8$, bound = 30, $T_{nd} = 100$ ms, and $k \cdot \text{coh}$ ranged from 0.003 to 0.023]

Again it's not the intermediate (dynamic) conditions I'm worried about. I'd bet 100 bucks that those will still strongly support the double integration model, especially given the 'concavity' of the overall pattern in Fig. 3a compared to convexity of my simulated data. Converting the dynamic evidence profiles to a presumed (latent) evidence strength that is fixed within a trial is a little tricky, since a simple average would have to specify up to what time. But assigning some stationary representation of momentary evidence (despite what's on the screen) is exactly what you'd want for a model that is explicitly designed as an alternative that does not care about dynamic evidence.

Anyway the details of the simulation shouldn't matter much — my only point is to demonstrate that a single accumulator can at least approximate the edge conditions, even with interleaving, and constrained to used the presented stimulus evidence (because that does not vary for the edge conditions!), without compromising parsimony.

What am I missing?

We thank the reviewer for this comment and agree it is important to fairly represent the classic model. We had included in the Supplementary Material (Figure S6) a model that roughly matches that described by the reviewer. The reviewer's prediction is correct that formal model comparison still strongly favours the double-integration model, and an important aspect of this is the concavity of the overall pattern of response times. We have now included this model in the main manuscript, which we call the 'classic with bottom-up weighting'.

Specifically, this model uses a weighting term (described as k by the reviewer) to scale the momentary evidence prior to accumulation (requiring one additional parameter). This weighting helps to spread the response times across conditions, and the accuracy range is also approximated. This model is not exactly as described by the reviewer because the stimuli don't have a 'coherence'. Coherence refers to the proportion of dots moving in one consistent direction (with other dots moving in random directions). Rather, the model takes the stimulus evidence based on the distribution of dot directions in our stimuli and overweights the variance of the dot directions sampled from the circular gaussian (using the same equation (9) as used to weight the computation of confidence in the double-integration model). In this way the model still describes behaviour based on what is presented on the screen. The model is also computationally close to the double-integration model, but importantly, it lacks the filtering in

the secondary integration stage. This demonstrates that a single accumulator can at least approximate the edge conditions, even with interleaving, and constrained to use the presented stimulus evidence, but this model does compromise on parsimony (with one additional parameter).

The models we compare therefore take the same input to predict the same responses, they also all rely on an evidence accumulation approach. What differs between the models is the functional architecture they are equipped with (whether there is this confidence control stage). The critical question is whether including online confidence control (via the secondary integration stage) is a better description of how human observers respond to this dynamic environment.

We choose to keep the classic model as is in the main manuscript because this model describes the first stage of evidence accumulation (which is then re-integrated in the double accumulation model). This first stage maps on to central parietal positivity in the EEG signals, commonly associated with evidence accumulation, while the secondary integration of the double-integration model is predicted by motor-sensor signals. The classic model with a bottom-up weighting scales the evidence such that the profile more closely matches that of the motor accumulator, and so is also more strongly predicted by EEG signals from sensors over motor cortex (Supplementary Figure S6).

Figure 3 in the main manuscript now presents the additional bottom-up weighting model:

Figure 3. Computational modelling. **a)** Participant behaviour (circular markers, black line) compared to that predicted by the classic single-integration model (square, red line). Colours correspond to **Figure 1b**, inset. **b)** Participant behaviour (circular markers, solid line) compared to the classic model with a bottom-up weighting of evidence accumulated (cross markers, red dashed line). **c)** Participant behaviour (circular markers, solid line) compared to the double integration model with online confidence control (asterisk markers, dashed line). **d)** Probability density of responses by response time. Correct responses are shown upward of the 0-point, and incorrect, downward. Markers show participants, red solid line shows the prediction of the classic model, red dashed line shows the prediction of the classic model with bottom-up weighting, and black dashed line shows the prediction of the confidence control model. **d)** Participant proportion correct (left) and median response time (right) across stimulus conditions, compared to that predicted by simulating the confidence control model using each participants' fitted parameters. **e)** Participants' confidence (z-scored) by confidence predicted from the model simulated confidence (using parameters fit to choices and response times but not confidence), across trials split by median response time (fast RT vs slow RT, left) and correct/incorrect discrimination decisions (right).

Which is described in the text alongside the original classic model:

“Our main hypothesis was that incorporating confidence control in the double integration framework would provide a better description of participant behaviour than the classic single-integration model. The classic single-integration model dramatically failed to capture the pattern of behaviour across intermixed stimulus conditions (**Figure 3a**). The model was unable to describe both how participants could be faster and more accurate with early high-quality evidence while also slowing down with early low-quality evidence. This failure is in part due to the fact that the model has to implement a single decision bound across the intermixed stimulus conditions, and in part due to constraining all models to use the evidence from the presented stimuli (a constraint validated by a separate GLM analysis, **Supplementary Figure S1**). This second constraint can be somewhat ameliorated by adding a bottom-up weighting of the evidence as it is accumulated (**Figure 3b**), to weaken the influence of more variable stimulus evidence (an overcompensation in the computation of decision evidence; see **Equations 1-3, 11 in Methods**). While this bottom-up weighting improves the spread of response times across conditions, the model overestimates performance, and the additional parameter used to implement the weighting is not parsimonious (bottom-up weighting – classic $\sum \Delta BIC = 38.14$; the model also suffers a lack of neurobiological plausibility; see **Supplementary Figure S6**).”

We also include model comparison with the double-integration model:

“Formal model comparison suggested the double-integration model with online confidence control proved vastly superior to the classic model (**Figure 3C**; $\sum BIC_{classic} = 8.33 \times 10^4$, $\sum BIC_{confcontrol} = 8.25 \times 10^4$, $\sum \Delta BIC = 842.61$, protected exceedance probability = 0.981; see **Supplementary Table S4** for a summary of parameters and fit statistics) and the classic model with bottom-up weighting ($\sum BIC_{bottom-up} = 8.34 \times 10^4$, $\sum \Delta BIC = 880.47$, protected exceedance probability = 0.989).”

And the relevant equation in the methods:

“As an additional check, we examined whether a weighting on the variability of the decision evidence could explain behaviour in a bottom-up manner, that is, affecting the primary evidence as it is accumulated in a single-integration framework (**Figure 3b**). We implemented this by applying the weighting described by Equation 9 on the primary evidence as it is accumulated.

$$x_{L,t} = x_{L,t-1} + \frac{\beta}{\beta + \sigma_t^2} v_{L,t} + \varepsilon_{L,t} \quad (11)$$

A side note: even if we ignore the additional complexity of the bottom-up weighting model, the double-integration model still proves superior in terms of the negative log likelihood: $\sum NLL_{bottom-up} = 4.14 \times 10^4$; $\sum NLL_{confcontrol} = 4.09 \times 10^4$, $\sum \Delta NLL = 440.37$; the median $\Delta NLL = 20.15$; all but three participants showed superior NLL for the confidence control model, for the three who did not the average $\Delta NLL = -9.63$.

We thank the reviewer for following up on this point, we hope that the inclusion of this additional model in the main manuscript helps to more fairly represent the classic model.

REVIEWERS' COMMENTS

Reviewer #2 (Remarks to the Author):

Great, thanks for the explanation. I did see the bottom-up weighting model in Suppl but did not appreciate the connection to 'k' in my version of the classic model. Still not sure I totally understand it, but the revisions are a more-than-reasonable compromise, and make the important point that even if the ranges are matched, the double integration model nails the shape much better.

No further comments; congrats on another cool study.